# The complete assembly of human LAT1-4F2hc complex provides insights into its regulation, function and localisation

Di Wu [1,2] ✉, Renhong Yan [3], Siyuan Song [1,2], Andrew K. Swansiger [4], Yaning Li[5], James S. Prell [4], Qiang Zhou [6] & Carol V. Robinson [1,2] ✉

The LAT1-4F2hc complex (SLC7A5-SLC3A2) facilitates uptake of essential amino acids, hormones and drugs. Its dysfunction is associated with many cancers and immune/neurological disorders. Here, we apply native mass spectrometry (MS)-based approaches to provide evidence of super-dimer formation (LAT1-4F2hc)$_2$. When combined with lipidomics, and site-directed mutagenesis, we discover four endogenous phosphatidylethanolamine (PE) molecules at the interface and C-terminus of both LAT1 subunits. We find that interfacial PE binding is regulated by 4F2hc-R183 and is critical for regulation of palmitoylation on neighbouring LAT1-C187. Combining native MS with mass photometry (MP), we reveal that super-dimerization is sensitive to pH, and modulated by complex N-glycans on the 4F2hc subunit. We further validate the dynamic assemblies of LAT1-4F2hc on plasma membrane and in the lysosome. Together our results link PTM and lipid binding with regulation and localisation of the LAT1-4F2hc super-dimer.

Heteromeric amino acid transporters (HATs) are a class of solute carrier (SLC) composed of a heavy subunit (SLC3 family) and a light subunit (SLC7 family). The light chain, featuring 12 transmembrane domains, exerts the amino acid transport function, whereas the glycosylated heavy chain is thought to regulate the heterodimer subcellular localisation and fine-tune its transport activity[1]. The L-type amino acid transporter 1 (LAT1, SLC7A5) is one of the most important HATs since it plays a pivotal role in facilitating uptake of large neutral amino acids, hormones and drugs across the membrane[2]. LAT1 forms a heterodimer with the 4F2 cell-surface antigen heavy chain (4F2hc, SLC3A2) via a disulfide bond between LAT1-C210 and 4F2hc-C164 (Fig. 1A). Interestingly, the regulatory roles of 4F2hc on the transport activity, selectivity and plasma membrane residence of LAT1 are not fully understood, given the conflicting findings obtained from in vitro and in vivo studies[3-7]. The LAT1-4F2hc (defined here as the heterodimer) is however widely expressed in various tissues, including leukocytes, pancreatic β-cells, and the blood-brain barrier. When aberrantly expressed, LAT1-4F2hc mediates the uptake of leucine and tunes the growth of tumour cells through the mechanistic target of rapamycin (mTOR) pathway, implicating the transporter in several human malignancies[2,8]. Understanding the regulation of the LAT1-4F2hc heterodimer is therefore crucial for drug development and therapeutic delivery to tumours and the brain[9,10].

The LAT1-4F2hc heterodimer is extensively decorated with post-translational modifications (PTMs), reported as four N-glycans on 4F2hc and several potential phosphorylation sites (Fig. 1B). The N-glycosylation status of the LAT1-4F2hc heterodimer is related to its plasma membrane residence and interactions with other transporters

[1]Department of Chemistry, University of Oxford, Oxford OX1 3QZ, UK. [2]Kavli Institute for Nanoscience Discovery, University of Oxford, Oxford OX1 3QU, UK. [3]Department of Biochemistry, Key University Laboratory of Metabolism and Health of Guangdong, School of Medicine, Southern University of Science and Technology, Shenzhen 518055 Guangdong Province, China. [4]Department of Chemistry and Biochemistry, 1253 University of Oregon, Eugene, Oregon 97403-1253, USA. [5]Beijing Advanced Innovation Center for Structural Biology, Tsinghua-Peking Joint Center for Life Sciences, Tsinghua University, Beijing 100084, China. [6]Research Center for Industries of the Future, Zhejiang Key Laboratory of Structural Biology, School of Life Sciences, Westlake University; Institute of Biology, Westlake Institute for Advanced Study; Westlake Laboratory of Life Sciences and Biomedicine, Hangzhou 310024 Zhejiang Province, China. ✉e-mail: di.wu2@chem.ox.ac.uk; carol.robinson@chem.ox.ac.uk

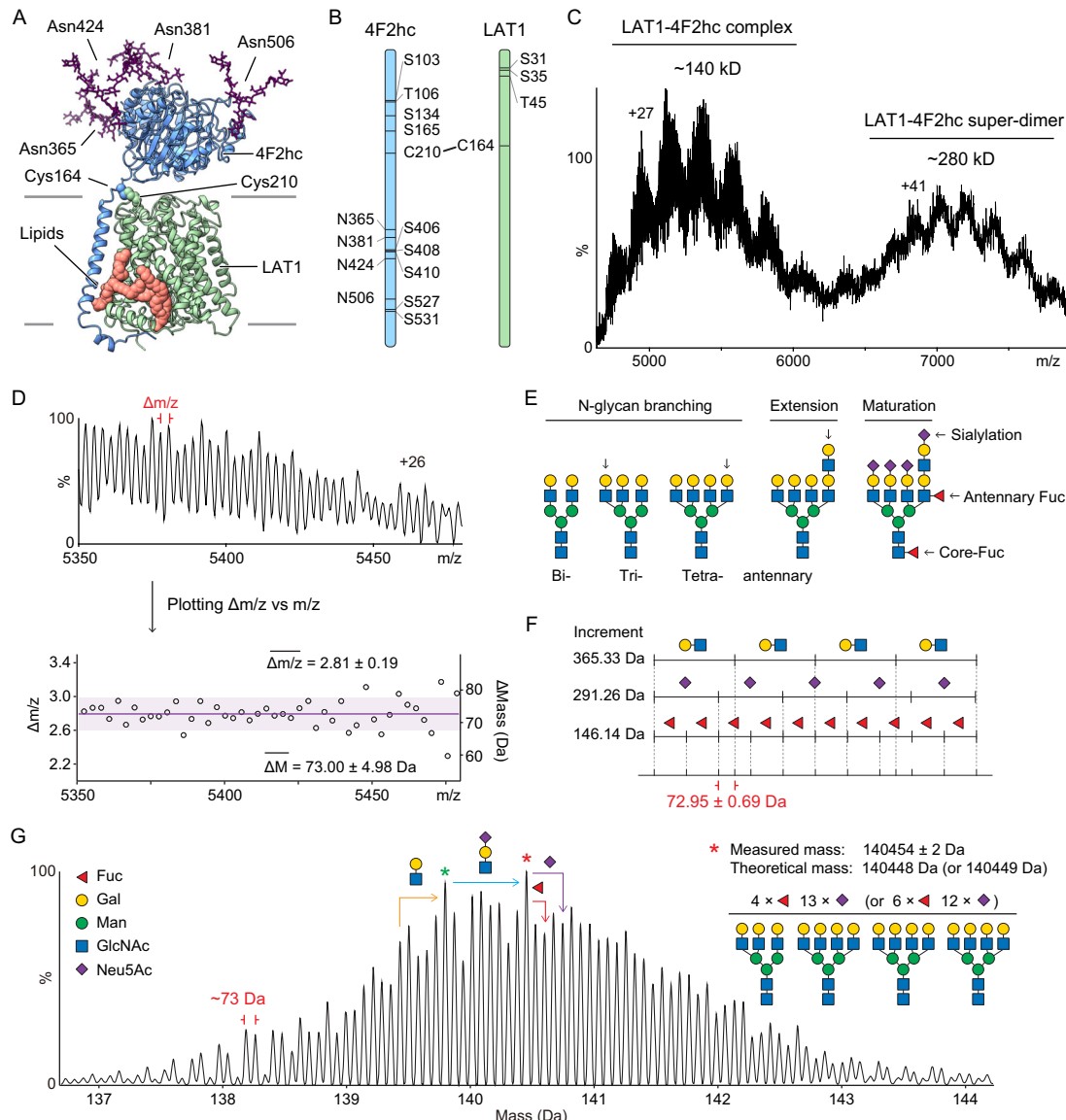

**Fig. 1 | MS analysis of the LAT1-4F2hc complex. A** Structure of LAT1-4F2hc as reported previously (PDB: 6IRT). Four N-glycans at Asn365, Asn 381, Asn424 and Asn506 are highlighted in purple. The covalently linked 4F2hc-Cys164 (blue) and LAT1-Cys210 (green) are shown. Two lipid-like molecules are shown (pink space filled). **B** Annotation of potential PTM sites, namely phosphorylation, N-glycosylation and disulfide bonds on 4F2hc and LAT1. **C** Mass spectrum of the LAT1-4F2hc complex from OGNG micelles. The molecular masses of heterodimeric and an unexpected super-dimeric form of the LAT1-4F2hc complexes are calculated as ~140 kDa and ~280 kDa, respectively. **D** An expansion of the +26 charge state from the native mass spectrum of the heterodimeric LAT1-4F2hc complex. The Δm/z between the adjacent peaks is calculated and plotted as a scatter plot. The averaged Δm/z (2.81) is plotted as a purple line with the shaded area corresponding

to one standard deviation (±0.19, light purple). Each fine structure peak differs by ~73 Da. **E** Illustration of anticipated N-glycan modifications on LAT1-4F2hc complex. N-glycan branching, extension, sialylation and fucosylation are common features found in the glycome of LAT1-4F2hc. **F** An illustration of how the combinatorial effects of N-glycan branching, extension, sialylation and fucosylation impact the molecular weight (MW) of the intact glycoprotein. Different combinations of these monosaccharide residues result in repeating 72.95 ± 0.69 Da increments on the intact glycoprotein masses (Supplementary Fig. 2). **G** Zero-charged spectrum of LAT1-4F2hc. The base peak labelled with a red asterisk is assigned as LAT1-4F2hc with one tri-antennary and three tetra-antennary sialylated N-glycans with four or six additional fucose residues. The N-glycan compositions of other peaks can be inferred based on the mass differences to the base peak.

through the galectin-lattice[11]. Recent structural studies reported a conserved phospholipid binding site at the interface of this, and other HAT heterodimers, namely LAT1-4F2hc, LAT2-4F2hc, xCT-4F2hc and b[0,+]AT-rBAT complexes (Fig. 1A and Supplementary Fig. 1)[12–16]. The 4F2hc-R183 site, which is in close vicinity to this phospholipid, is critical to transport activity (Supplementary Fig. 1)[12]. Moreover LAT1, as one of the LeuT-fold amino acid transporters, is proposed to be assembled as a dimer in the membrane[17], while rBAT (SLC3A1), a 4F2hc homologous heavy chain, has also been shown to mediate super-dimerization of HAT heterodimers[15,18]. The super-dimerization status of the LAT1-4F2hc heterodimer and its regulation through its

heterogeneous N-glycosylation repertoire, endogenous phospholipids and PTM status, however, remain challenging to study via established structural biology approaches.

Recent advances in native mass spectrometry (nMS) enable definition of the heterogeneity and regulation of soluble protein assemblies[19]. However, for membrane protein complexes, the situation is further complicated by the co-existence of additional endogenous PTMs and lipid interactions together with the problem of residual detergent molecules adhering to the protein in the gas phase. Collectively these factors combine to yield complicated spectra with many overlapping peaks[20]. To overcome this complexity, here we develop

and apply a high-resolution native MS approach to probe the assembly of the LAT1-4F2hc complex released from detergent micelles, and reveal two endogenous PE lipids binding to the C-terminus of each LAT1 subunit. Notably, we show that 4F2hc-R183 is the key residue for maintaining endogenous PE at the heterodimeric interface. Moreover, we unveil four highly branched N-glycans with 9 to 16 sialic acid residues on each 4F2hc subunit within each LAT1-4F2hc heterodimer. We further show how the complete assembly with four N-glycans, particularly the terminal sialic acid residues, mediate super-dimerization of the LAT1-4F2hc complex.

## Results

### Heterogeneous glycosylation of intact LAT1-4F2hc complexes

We expressed fully glycosylated LAT1-4F2hc complexes in HEK293F cells and purified the protein complexes in a mild non-ionic detergent, digitonin, following an established protocol used for the Cryo-EM study of LAT1-4F2hc[12]. We first recorded a mass spectrum on a Q Exactive-UHMR of the LAT1-4F2hc complex in glyco-diosgenin (GDN), a synthetic substitute for digitonin. However, the mass spectrum of LAT1-4F2hc shows unresolved signal, spanning from 6000 to 20000 m/z, suggesting the presence of lipids and detergent remaining bound to the protein complexes in the gas phase, even at high activation energies[21] (Supplementary Fig. 1E). Therefore, we performed a detergent screen by buffer-exchanging the LAT1-4F2hc complex into different detergents[22]. Notably, the LAT1-4F2hc complex was successfully released from octyl glucose neopentyl glycol (OGNG) micelles (Fig. 1C). The resulting mass spectrum is consistent with extensive glycosylation, evidenced by the broadening and splitting of charge state series. Two distributions of charge states could be assigned to LAT1-4F2hc heterodimer and a dimer of heterodimers (LAT1-4F2hc)$_2$ (defined here as the super-dimer) with molecular masses of 140 kDa and 280 kDa respectively (Fig. 1C). The LAT1-4F2hc heterodimer peaks exhibit higher resolution than the super-dimeric ones with a repeating interval (~73 Da), implying that this peak splitting / fine structure could arise from resolved proteoforms (Fig. 1D).

To understand the origin of these unidentified proteoforms we considered first N-glycosylation. For mammalian proteins, N-glycosylation, fucosylation (addition of Fuc residue; Fuc: fucose), sialylation (addition of Neu5Ac residue; Neu5Ac: N-acetylneuraminic acid) and N-glycan branching (addition of GlcNAc$_1$Gal$_1$; GlcNAc: N-acetylglucosamine; Gal: galactose) are the three main features that contribute to N-glycoprotein heterogeneity (Fig. 1E). Addition of repeating Fuc, Neu5Ac and GlcNAc$_1$Gal$_1$ residues to an intact glycoprotein results in intervals with a 72.95 ± 0.69 Da difference between each proteoform (Fig. 1F and Supplementary Fig. 2). A manual examination of the fine structure on LAT1-4F2hc confirmed this repeating interval on all charge states, suggesting that these peaks are due to resolved glycoforms with different monosaccharide composition (Fig. 1D and Supplementary Fig. 2C).

Having established this mass interval, we were able to deconvolve the native mass spectrum using two software approaches (i) Bayesian probability-based UniDec[23] and (ii) Fourier transform-based iFAMs[24] (mass filter of 72.95 Da used in both cases) (Fig. 1G and Supplementary Fig. 3). Similar results were obtained using both approaches, allowing us to tentatively assign the most abundant proteoform, with a measured mass of 140454 ± 2 Da, to LAT1-4F2hc with HexNAc$_{23}$Hex$_{27}$Fuc$_4$Neu5Ac$_{13}$ or HexNAc$_{23}$Hex$_{27}$Fuc$_6$Neu5Ac$_{12}$ (red asterisk, Fig. 1G; Hex, hexose; HexNAc, N-acetylhexosamine). This assignment is based on the measured mass and the previous glycomics study of the 4F2hc subunit[11].

The possible glycan composition of other peaks can be inferred based on mass differences of 146.14 Da for fucosylation, 291.26 Da for sialylation and 365.33 Da for N-glycan branching (Fig. 1G). For example, the measured mass of 139797 ± 2 Da (green asterisk) can be assigned either to HexNAc$_{22}$Hex$_{26}$Fuc$_4$Neu5Ac$_{12}$ or HexNAc$_{22}$Hex$_{26}$Fuc$_6$Neu5Ac$_{11}$

(Fig. 1G). Notably, two GlcNAc$_1$Gal$_1$ units (730.67 Da) or five Fuc residues (730.71 Da), and one Neu5Ac (291.26 Da) or two Fuc residues (292.28 Da) cannot be differentiated at the intact glycoprotein level with native MS. However, we can use the LAT1-4F2hc proteoforms to simulate the super-dimer heterogeneity using a binomial model[25] (Supplementary Fig. 4). Using this approach, of matching simulated and raw data, we found that the theoretical average mass of the LAT1-4F2hc super-dimer (calculated from two times the measured mass of the heterodimer (280954 Da) agrees well with the MS-measured value (280934 ± 109 Da). Our mass measurements, therefore imply that the super-dimer does not require additional interfacial lipids or ligands, compared to the LAT1-4F2hc heterodimer, to maintain its super-dimeric status.

### Reducing the glycan heterogeneity of the LAT1-4F2hc complex

To elucidate the proteoforms and co-purified lipids in LAT1-4F2hc complexes, we simplified the heterogeneity of the fully glycosylated complex using neuraminidase digestion (Fig. 2A). This treatment removes sialylation but retains fucosylation and N-glycan branching such that these two features become the primary source of glycan heterogeneity. Repeating our native MS experiments for desialylated LAT1-4F2hc heterodimers, the peak splitting is now readily resolved with a major peak series P1 – P10 (differing by 365.33 Da) assigned to GlcNAc$_1$Gal$_1$ units (Fig. 2B and C). For example, we attributed the P5 proteoform, with a measured mass of 137027 ± 2 Da, to the LAT1-4F2hc heterodimer with four core-fucosylated tetra-antennary N-glycans (HexNAc$_{24}$Hex$_{28}$Fuc$_4$), an assignment supported by glycoproteomics analysis of 4F2hc at site-specific level (Fig. 2C and Supplementary Fig. 5A). Furthermore, we assigned other less abundant proteoforms (differing by 146 Da) to glycoforms with one to four antennary Fuc residues (Fig. 2C, highlighted with aF1 to aF4). Interestingly, tri- and tetra-fucosylated forms are more abundant than bi-fucosylated species. This finding differs from previous observations of highly fucosylated glycoproteins wherein intensity reduces with the number of antennary fucosylation events[26–28] (Fig. 2D). Notably, we found evidence of an additional GlcNAc residue at Asn365, and two phosphorylation sites, one on both of the 4F2hc and LAT1 subunits, via bottom-up proteomics analysis (Supplementary Fig. 5A to C). The overlapped tri-fucosylated P3 proteoform (P3, aF3) and phosphorylated P4 proteoform (P4, aF0) result in a mass difference of 74.0 ± 0.7 Da (Fig. 2D and Supplementary Fig. 5D). Similarly, the mass difference of 214.7 ± 2.2 Da could be attributed to the overlapped tetra-fucosylated P3 proteoform (P3, aF4) and P4 proteoform (P4, aF0) with GlcNAc.

Having annotated all desialylated LAT1-4F2hc heterodimer peaks with different levels of N-glycan branching and fucosylation, we found that the fully sialylated LAT1-4F2hc heterodimer carries 9 to 16 sialic acid residues, confirming the annotation of the sialylated LAT1-4F2hc heterodimer (Supplementary Fig. 6 and Fig. 1F). However, we were still not able to clearly distinguish lipid-bound peaks in this highly heterogeneous spectrum. It is worth noting that the P1 – P10 peaks can be fitted with two partially overlapped Gaussian curves (Supplementary Fig. 6A). Since such a distribution does not result from a typical distribution of glycosylated proteoforms, this observation implies that at least one lipid-bound peak may overlap with other proteoforms, namely the P5 – P10 series, and contribute to the second Gaussian envelope.

To confirm the presence of phospholipids in this LAT1-4F2hc complex from a human cell line, we performed an MS-based lipidomics experiment to identify co-purified phospholipids. We identified six phospholipid species associated with the complex with masses spanning from 670 Da to 920 Da, namely phosphatidic acid (PA), phosphatidylethanolamine (PE), phosphatidylglycerol (PG), phosphatidylcholine (PC), phosphatidylserine (PS) and phosphatidylinositol (PI) (Fig. 2E). Moreover, we also identified cholesterol in co-purified lipids (Supplementary Fig. 7), in line with the previous observation[12,13]. Since binding of cholesterol to the intact heterodimer is not expected to survive the transition to the gas phase, we

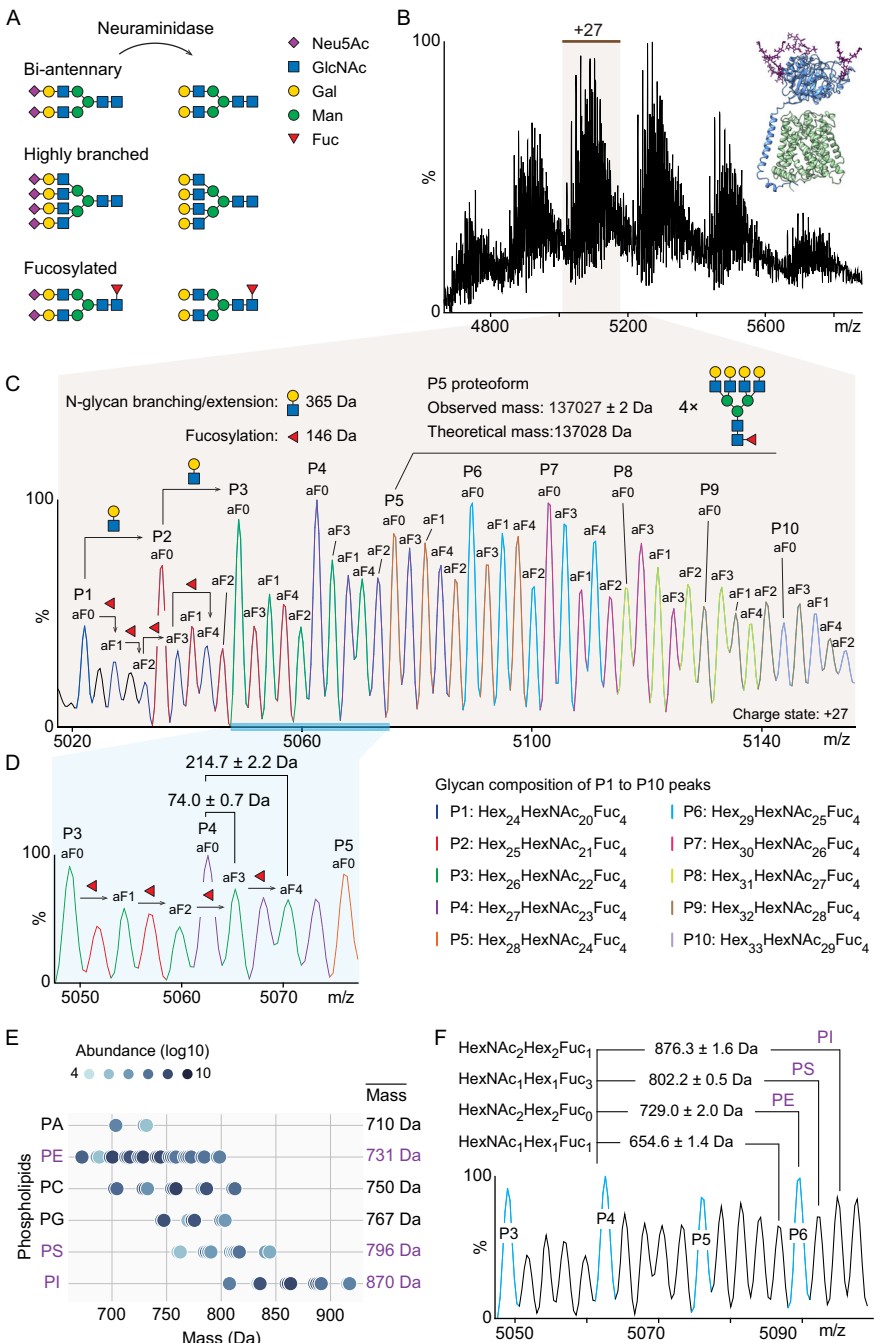

**Fig. 2 | MS analysis of the desialylated LAT1-4F2hc complex. A** Schematic illustration of neuraminidase treatment of N-glycans in which all sialic acid residues are removed from N-glycans. **B** Native mass spectrum of desialylated LAT1-4F2hc. **C** Annotation of the N-glycan heterogeneity on desialylated LAT1-4F2hc (charge +27). The major peaks, differing by one $GlcNAc_1Gal_1$ unit (365.33 Da), are labelled with P1 – P10. The peaks carrying zero to four antennary fucose residues are annotated with aF0 to aF4, respectively. The P5 proteoform (measured mass of 137027 ± 2 Da) was assigned to LAT1-4F2hc complex with four tetra-antennary core-fucosylated N-glycans (theoretical mass of 137028 Da). **D** Annotation of antennary fucosylation status of P3 proteoforms. The P3 aF3 and aF4 proteoforms are 74.0 ± 0.7 Da and 214.7 ± 2.2 Da larger than the P4 aF0 proteoform, respectively. **E** Lipidomics analysis of co-purified phospholipids with LAT1-4F2hc. The identified phospholipids, namely PA, PE, PC, PG, PS and PI are plotted according to their relative abundances (light to dark blue dots). **F** Annotation of the peaks of the co-purified lipids (PE, PS and PI) with mass shifts (650–900 Da) and their potential overlap with glycoforms of the P4 proteoform.

considered only the phospholipids. Notably, the weighted average masses of PE, PS and PI are 731 Da, 796 Da and 870 Da, respectively. These masses may therefore overlap proteoforms with additions of $GlcNAc_2Gal_2$ (730.3 Da), $GlcNAc_1Gal_1Neu5Ac_1Fuc_1$ (802.3 Da) and $GlcNAc_2Gal_2Fuc_1$ (876.3 Da), respectively (Fig. 2F and Supplementary Fig. 7A). This overlap prevents us from extracting more information about the detailed phospholipid species present and their binding stoichiometry from this desialylated complex. Therefore, we decided

to reduce the glycan complexity yet further by dissociating the LAT1-4F2hc complex in the gas phase and probing lipid binding to the LAT1 and 4F2hc subunits separately.

## Resolving endogenous lipid binding to the LAT1-4F2hc heterodimer

We first performed gas-phase dissociation of the intact desialylated LAT1-4F2hc heterodimer with increased collisional energy

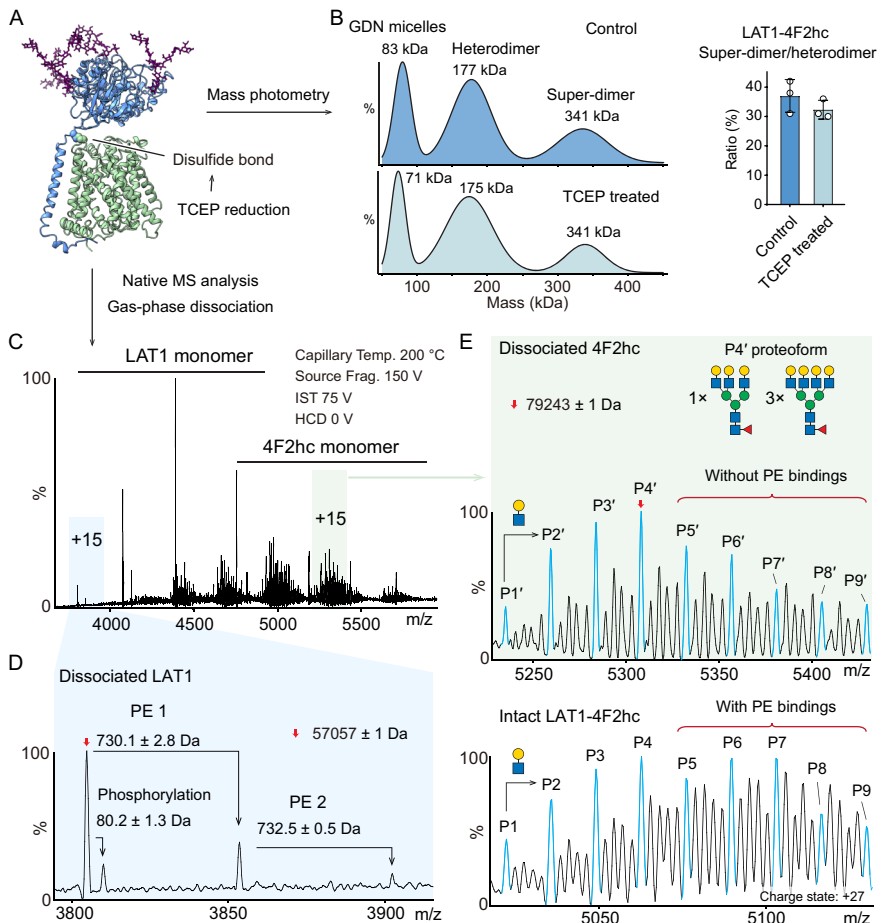

**Fig. 3 | Native MS analysis reveals endogenous PE binding to the dissociated LAT1-4F2hc complex. A** TCEP treatment reduces the disulfide bond between LAT1 and 4F2hc subunits. **B** Mass photometry measurements of LAT1-4F2hc assemblies in 36 μM GDN without (control) and with TCEP treatment. The molecular weights of LAT1-4F2hc heterodimer and super-dimer in GDN proteomicelles are ~177 kDa and 341 kDa, respectively. The bar graph shows ratios of LAT1-4F2hc heterodimer and super-dimer without (control) and with TCEP treatment. Bars show mean ± standard deviation from three independent experiments (dots). Source data are provided as a Source Data file. **C** Native MS analysis of the TCEP-treated LAT1-4F2hc and gas-phase dissociation of the two subunits. The native MS parameters, including capillary temperature (Capillary Temp.), source fragmentation energy (Source Frag.), In-source trapping energy (IST) and HCD energy (HCD) are labelled. **D** The spectrum of dissociated LAT1 (charge state + 15) reveals one phosphorylation (80.2 ± 1.3 Da) and two lipid adducts (730.1 ± 2.8 Da and 732.5 ± 0.5 Da). **E** The spectra of dissociated 4F2hc subunit (charge state + 15, top panel) and desialylated LAT1-4F2hc complex (charge state + 27, bottom panel). The corresponding proteoforms of the 4F2hc subunit (P1′ – P9′) and the LAT1-4F2hc heterodimer (P1 – P9) are aligned. The absence of the LAT1 subunit and its associated endogenous lipids results in the decreased abundances of P5 to P9 peaks of the dissociated 4F2hc subunit (the upper spectrum highlighted in green).

(Supplementary Fig. 8A). The LAT1 and 4F2hc subunits are covalently linked via a disulfide bond and interact with each other extensively at the heterodimeric interface[12]. The majority of LAT1-4F2hc heterodimers remain intact in the gas phase, however a small population of dissociated LAT1 and 4F2hc is observed following gas-phase activation and may be not covalently linked by the disulfide bond. However, the background is too high to annotate the adducts and proteoforms of each subunit, and the relatively high collision energy strips the bound lipid from LAT1 subunits (Supplementary Fig. 8B). We, therefore treated the LAT1-4F2hc heterodimer with Tris(2-carboxyethyl)phosphine (TCEP) to fully reduce the disulfide bond in solution, prior to gas phase dissociation (Fig. 3A). Probing the integrity of the TCEP-treated LAT1-4F2hc complex, first with mass photometry, we found that the heterodimer and super-dimer equilibrium of LAT1-4F2hc assemblies is not affected by TCEP reduction (Fig. 3B and Supplementary Fig. 8C). We then performed native MS to analyze the non-covalently linked LAT1-4F2hc complex. We observed dissociated LAT1 and 4F2hc subunits released from proteolipomicelles, with no evidence of the intact LAT1-4F2hc heterodimer remaining (Fig. 3C and Supplementary Fig. 8D). Importantly, we detected two phospholipid adduct peaks (730.1 ± 2.8 Da and

732.5 ± 0.5 Da) corresponding to two different phospholipids bound to the LAT1 monomer (Fig. 3D).

We compared the heterogeneity of the dissociated 4F2hc subunit to the proteoforms of the intact LAT1-4F2hc complex (Fig. 3E). The spectrum of dissociated 4F2hc subunits shows decreased abundances of the P5′ to P9′ proteoforms (Fig. 3E, top panel), relative to the peaks assigned to the LAT1-4F2hc heterodimer (Fig. 3E, bottom panel, P5 to P9 proteoform). This observation supports our conjecture that the lipid-bound species contributes to the increased abundances of the P5 to P9 series in the desialylated, intact LAT1-4F2hc heterodimer. As discussed above, the phospholipid-bound peaks overlap with the N-glycan branched peaks with two additional GlcNA₁Gal₁ units (730.67 Da). Based on the lipidomics analysis of co-purified lipids associated with LAT1-4F2hc (Fig. 2E), we attribute these adducts to PE with a weighted average mass of 731 Da.

After elucidating the head group of the endogenously bound lipid, we analyzed the hydrophobic chain of PE. Interestingly, lipidomics analysis of LAT1-4F2hc co-purified lipids identified two PE species, namely diacyl-PE and alkylacyl-PE that differ by one oxygen atom (Supplementary Fig. 9A). We, therefore performed a progressive delipidation experiment to differentiate the affinities of diacyl-PE and alkylacyl-PE to

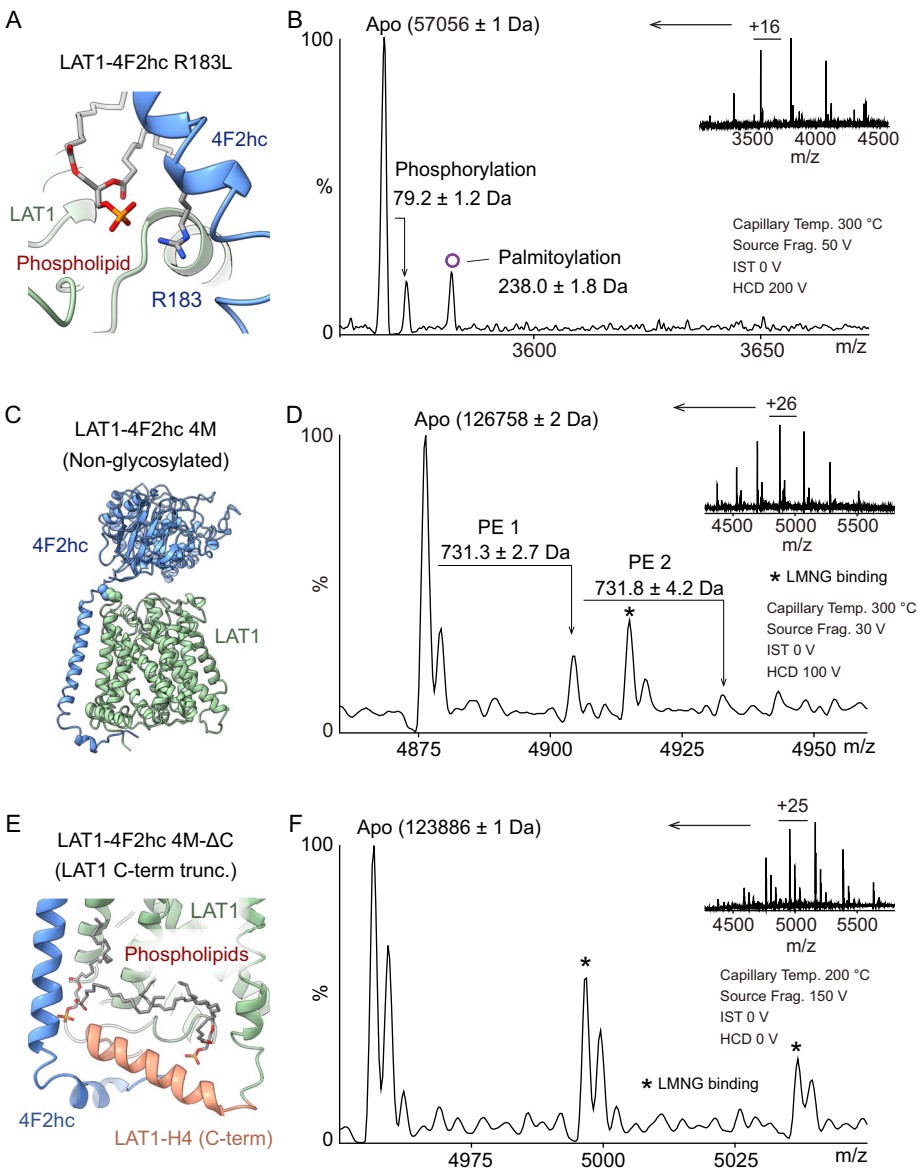

**Fig. 4 | Mutation of LAT1-4F2hc complexes to investigate lipid binding, the effects of glycosylation and the influence of the LAT1 C-terminus. A** The phospholipid binding site is at the C-terminus of the LAT1 subunit (light green) and the transmembrane segment of 4F2hc (blue). The head group of interfacial phospholipid interacts with 4F2hc-R183. **B** Native mass spectra of LAT1-4F2hc with the 4F2hc-R183L mutation. No lipid adduct is observed. The palmitoylated proteoform is labelled with a purple circle. **C** Structural illustration of non-glycosylated LAT1-4F2hc 4 M mutant with N365D/N381D/N424D/N506D. **D** Native mass spectra of non-glycosylated LAT1-4F2hc 4 M mutant in 20 μM LMNG. Two endogenous lipid adducts (731.3 ± 2.7 Da and 731.8 ± 4.2 Da) are observed. **E** Structural illustration of LAT1-4F2hc 4M-ΔC mutant with deletion of LAT1 C-terminal helix (highlighted orange). **F** Native mass spectra of LAT1-4F2hc 4M-ΔC mutant in LMNG. The absence of lipid adducts means that no endogenous lipids are retained. LMNG adduct peaks are highlighted with asterisks.

the LAT1-4F2hc complex[21] (Supplementary Fig. 9B). Interestingly, we found diacyl-PE, particularly diacyl-PE(C36:2) is enriched in our lipidomics analysis (Supplementary Fig. 9C). This suggests that diacyl-PE has a stronger binding affinity to LAT1-4F2hc than alkylacyl-PE.

**Locating the endogenous PE binding to LAT1-4F2hc complex**
Previously, we evidenced that the positively charged side chain of 4F2hc-R183 interacts with the phosphate group of a phospholipid using Cryo-EM[12] (Fig. 4A). Hence, we considered whether this phospholipid is the PE that we observed using native MS and lipidomics. We mutated this positively charged arginine at position 183 to neutral leucine and performed native MS analysis of the TCEP-treated mutant (4F2hc-R183L). We found that this mutation abolishes PE binding to the LAT1 subunit (Fig. 4B) allowing us to attribute PE to the lipid binding site at 4F2hc-R183L. Moreover, we found that this R183 mutation leads

to the appearance of a new adduct (+ 238 Da) compared to the wild type complex. This adduct, which can be removed through hydroxylamine treatment, is consistent with palmitoylation of LAT1 (Fig. 4B and Supplementary Fig. 10). Using a computational tool for the prediction of S-palmitoylation sites in proteins[29], LAT1-C187, in the vicinity of 4F2hc-R183, is predicted to be palmitoylated. These results lead us to propose that absence of the R183 lipid binding site, located close to the inner membrane bilayer, leads to palmitoylation suggesting that this region may play a role in efficient trafficking of the complex to the plasma membrane[30].

After identifying PE binding based on the gas-phase dissociated subunits, we further validated lipid binding within the intact LAT1-4F2hc heterodimer. We generated a non-glycosylated LAT1-4F2hc variant with four mutations (N365Q, N381Q, N424Q and N506Q; 4 M) to eliminate N-glycan heterogeneity (Fig. 4C). Since the 4 M mutant

does not electrospray well in OGNG, we analysed the mutant in lauryl maltose neopentyl glycol (LMNG) detergent. This spectrum confirmed retention of the two endogenous PE binding sites (731.3 ± 2.7 Da and 731.8 ± 4.2 Da) (Fig. 4D). Next, we generated a 4 M mutant with the LAT1 C-terminal residue 483-507 deleted (4M-ΔC) to further validate the lipid binding sites in the intact non-glycosylated complex (Fig. 4E). Native MS analysis of the 4M-ΔC mutant showed an absence of PE lipid binding, only detergent adducts remain (Fig. 4F). Together, these findings confirm that one PE lipid interacts at the heterodimer interface, interacting with 4F2hc-R183 as shown above, but that both are lost when the H4 helix of LAT1 at the C-terminus is removed.

Comparing our data with that obtained via Cryo-EM, we were able to model the PE molecules into the corresponding density. The head groups of both phospholipid molecules are well accommodated into the respective cryo-EM densities (Supplementary Fig. 11). The inter-facial PE headgroup interacts with LAT1-H367/Q501 and 4F2hc-R183, its hydrophobic tails inserting into the heterodimeric interface between LAT1-TM4/TM9/H4 and 4F2hc-TM (Supplementary Fig. 11C). The other PE resides in the hydrophobic pocket formed by LAT1 transmembrane helices LAT1-TM9, TM12 and H4, its head group interacting with LAT1-S361 (Supplementary Fig. 11D).

## LAT1 dimerization and PE binding are independent of phosphorylation and lipid binding status

LAT1 also carries multiple potential phosphorylation sites on its N-terminus (Fig. 1B), a PTM that has been reported to regulate solute carrier function[30]. We used native MS to observe a mono-phosphorylated proteoform and located the site to LAT1-S35 by means of proteomics analysis. (Fig. 4B and Supplementary Fig. 5C). This phosphorylation adds an extra negative charge to the flexible N-terminus of LAT1 (Supplementary Fig. 12A) which could have implications for dimerization and endogenous lipid recruitment[31]. To investigate these possibilities, we first analyzed the non-glycosylated LAT1-4F2hc 4 M mutant and compared its phosphorylation status with apo- and PE-bound forms of the complex (Supplementary Fig. 12B). We found that phosphorylation on LAT1-S35 does not significantly affect the extent of LAT1-PE binding. Next, we probed potential phosphorylation regulation of LAT1 homo-dimerization using native MS (Supplementary Fig. 12C). We quantified the relative abundance of phosphorylated LAT1 monomer and simulated the theoretical abundance of the phosphorylated LAT1 homo-dimer using a binomial distribution model, assuming that LAT1 homo-dimerization is independent of its phosphorylation status (Supplementary Fig. 12D). We found no significant difference between the theoretical and experimental relative abundances of phospho-proteoforms. Therefore, we conclude that phosphorylation of the N-terminus is not involved in lipid recruitment or LAT1 homo-dimerization.

We also considered the possibility that LAT1 dimerization is modulated by lipids in the membrane. This is an established mechanism for the prokaryotic homologous amino acid transporter LeuT, the leucine transporter[32]. The prokaryotic arginine-agmatine antiporter (Adic), another LeuT-fold transporter, can also form homodimers although the interface is distinct from that employed in LeuT (Supplementary Fig. 12E, F)[33,34]. Native MS of LAT1 reveals both monomeric and homodimeric protein complexes, importantly without any lipid binding to the monomer or homodimer (Supplementary Fig. 12C). This implies that LAT1 homodimerization is phospholipid-independent, differing from its prokaryotic homologue, LeuT[32] (Supplementary Fig. 12G). This result is further supported by the observation that the LAT1-4F2hc super-dimer is free of interfacial lipid when observed in native MS (Supplementary Fig. 13).

## N-glycan regulation of the LAT1-4F2hc super-dimerization

Considering further the super-dimerization of the complex, observed in native MS experiments (Fig. 1C), we note that another HAT, b[0,+]AT-

rBAT (SLC7A9-SLC3A1) forms a super-dimer mediated by the ectodomain (ECD) of rBAT[15] (Supplementary Fig. 14A). The super-dimerization of b[0,+]AT-rBAT is critical for its plasma membrane localisation and its transporter function[18]. We hypothesize that the ECD of 4F2hc, rather than LAT1, may mediate super-dimerization of LAT1-4F2hc. To investigate this possibility, we examined the 4F2hc subunit and simulated possible N-glycan conformers on a single LAT1-4F2hc heterodimer. We found that the four N-glycans conformers shield large surface areas of the 4F2hc subunit (Fig. 5A). This implies that the N-glycans may also be involved in super-dimer interactions. Indeed, a recent study suggested that N-glycosylation on 4F2hc is essential for its interactions with other proteins, such as membrane transporters and the galectin-lattice[11].

To determine experimentally whether glycans are implicated in super-dimer formation we used native MS to compare the relative abundance of LAT1-4F2hc heterodimer and super-dimer for the following complexes (i) WT fully glycosylated, (ii) desialylated and (iii) the non-glycosylated (4 M) mutants (Fig. 5C). We found that super-dimerization is most prevalent for the fully glycosylated WT, reduced after trimming the terminal sialic acids and not detected after removing all N-glycans. To support our native MS data, we then carried out solution-based mass photometry (MP) of WT and desialylated LAT1-4F2hc. A clear reduction of the dimer population from the WT to the desialylated complex is observed via MP measurements further validating the stabilization effect of terminal sialic acid residues observed in our MS data of LAT1-4F2hc super-dimerization (Fig. 5D, E).

Since LAT1-4F2hc activates the mTORC1 pathway upon the translocation of the complex from the plasma membrane into the lysosome[35], we reasoned that the acidic environment in the lysosome (pH 4.5–5) could also affect super-dimer formation. To investigate this, we first considered the effects of changing the surface charge of the 4F2hc interfacial area from negative to positive by adjusting the solution pH from 7.4 – 5.0 (Fig. 5F). As the sialic acid remains negatively charged at pH 5.0, the interactions between the positively charged 4F2hc interface and sialic acids may enhance super-dimerization of LAT1-4F2hc. Therefore, we monitored super-dimerization of LAT1-4F2hc at pH 5.0 using MP in GDN micelles, as the partially resolved super-dimer could not be accurately quantified at acidic pH in native MS[36]. We found a higher relative ratio of LAT1-4F2hc super-dimer to heterodimer in the acidic environment (Fig. 5D, E). Together, these results suggest that N-glycosylation of 4F2hc promotes LAT1-4F2hc super-dimer formation, a process that is favoured at low pH.

We used homology-based modelling of the b[0,+]AT-rBAT complex to model the LAT1-4F2hc super-dimer (Fig. 5B and Supplementary Fig. 14B). From our model we found that the 4F2hc subunit, which is smaller than rBAT, has two N-glycans on 4F2hc (Asn365 and Asn424) positioned such that they can interact extensively across the interface of the neighbouring 4F2hc subunits (Fig. 5B and Supplementary Fig. 14C). This is in contrast to the b[0,+]AT-rBAT complex where the two rBAT subunits contact each other to facilitate super-dimerization. In addition to our homology-based model, we generated AlphaFold-Multimer[37] models for the LAT1-4F2hc super-dimer (Supplementary Fig. 14D). Interestingly the AF models also show that N-glycans on Asn365 and Asn424 might facilitate LAT1-4F2hc super-dimerization. Moreover, we found that the 4F2hc attachment serves to disrupt LAT1 homodimerization due to clashes between the ECDs of each 4F2hc subunit in AF models and the homology model above (Supplementary Fig. 15). Our models are not however consistent with the LAT1 homo-dimer interface participating in LAT1-4F2hc super-dimerization.

## Probing endogenous LAT1-4F2hc assemblies from human cell lines

After elucidating the stoichiometry of LAT1-4F2hc assemblies in vitro, and their response to glycosylation, phosphorylation, palmitoylation and lipid binding, we considered endogenously expressed LAT1-4F2hc

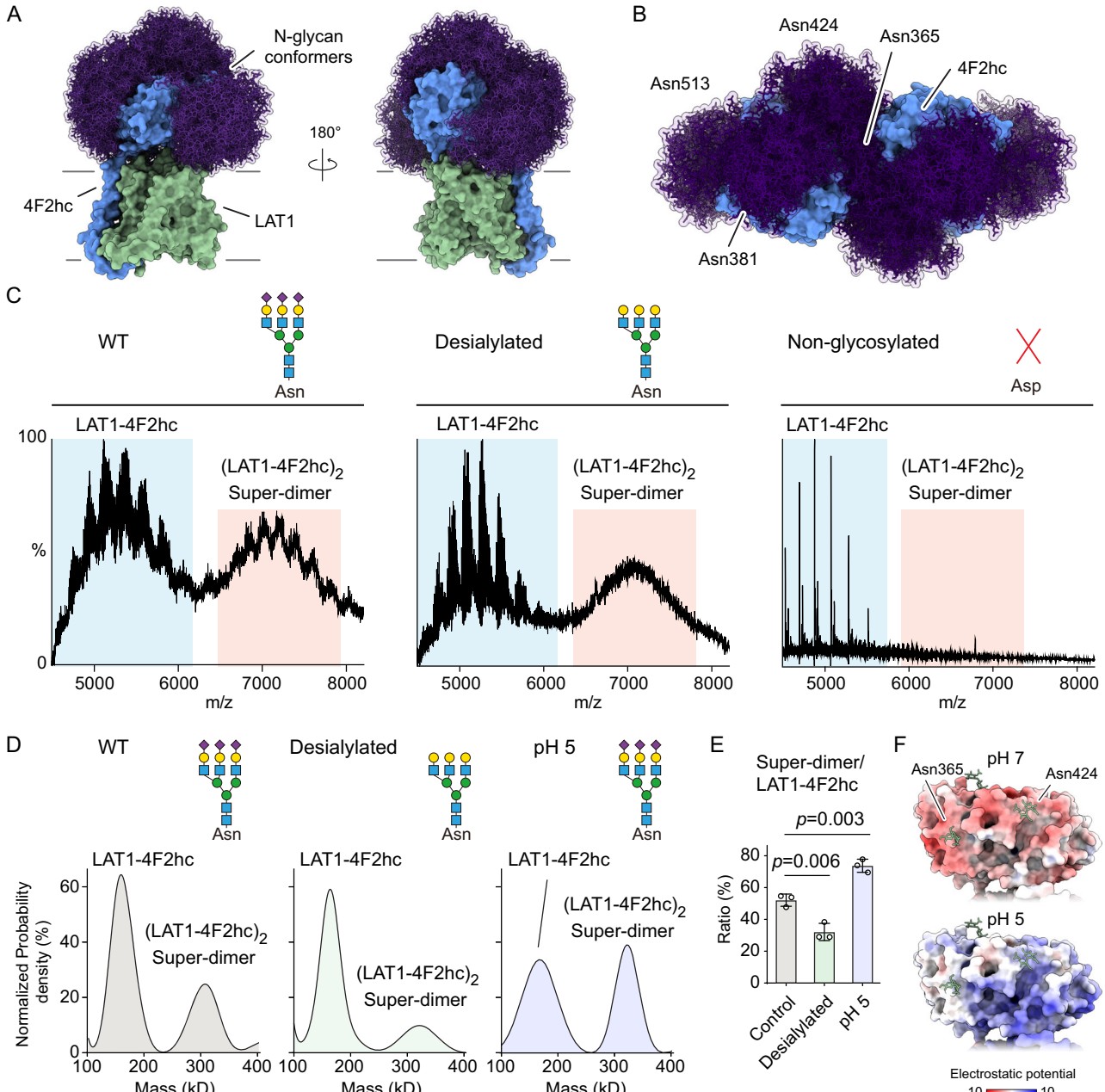

**Fig. 5 | Native MS and mass photometry analysis of LAT1-4F2hc super-dimerization. A** Simulation of possible N-glycan conformers on LAT1-4F2hc. N-glycan conformers are highlighted in dark purple. **B** Structural illustration of the modelled structure of LAT1-4F2hc super-dimer with possible N-glycan conformers. The 4F2hc-Asn365 and Asn424 are proximal to the super-dimer interface. The N-glycans fill the interface and mediate hydrophilic interactions between two 4F2hc subunits in the LAT1-4F2hc super-dimer. **C** Native mass spectra of LAT1-4F2hc WT, desialylated WT and non-glycosylated M4 mutant. The LAT1-4F2hc heterodimer and its super-dimer peaks are highlighted (blue and red, respectively). **D** Mass photometry analysis of LAT1-4F2hc fully sialylated WT (control), desialylated WT and sialylated WT at pH 5.0. **E** Bargraphs show the super-dimer/heterodimer ratios of fully sialylated WT (control), desialylated WT and sialylated WT at pH 5.0. Bars show mean ± standard deviation from three independent experiments (dots). Two tailed Student's $t$-tests are performed for statistical analysis ($p < 0.01$ was labelled with two asterisks). Source data are provided as a Source Data file. **F** Surface electrostatic potential of the super-dimer interface of 4F2hc subunit at pH 7.0 and 5.0.

assemblies. Firstly, we validated super-dimerization of endogenous LAT1-4F2hc in HeLa cells. We incubated the cells with BS3 (bissulfosuccinimidyl suberate) to crosslink cell surface LAT1-4F2hc complexes and probed their super-dimerization status with Western blotting (Fig. 6A). We found that LAT1-4F2hc forms a large complex above 200 kDa on HeLa cell membrane. We further affinity-purified crosslinked LAT1-4F2hc complexes using anti-LAT1 antibody and analysed the band above 200 kDa (Supplementary Fig. 16). We only identified LAT1 and 4F2hc subunits as the most abundant proteins (beside IgG) in this band. This result implies that the large crosslinked complex is the

LAT1-4F2hc super-dimer (Fig. 6A). This result supports that LAT1-4F2hc forms super-dimer in vivo.

Furthermore, we were interested in the ratio of protein expression of LAT1 and 4F2hc since this would inform the possible existence of free LAT1 in the membrane. Since the protein expression levels of 4F2hc (SLC3A2) and LAT1 (SLC7A5) are correlated with their mRNA levels (Supplementary Fig. 17A), we compared the mRNA levels of SLC3A2 and its associated SLC7A family members (SLC7A5, A6, A7, A8, A10, A11) in 50 human tissues and 1206 cell lines from the ProteinAtlas database[38] (measured in normalized transcript per million, nTPM)

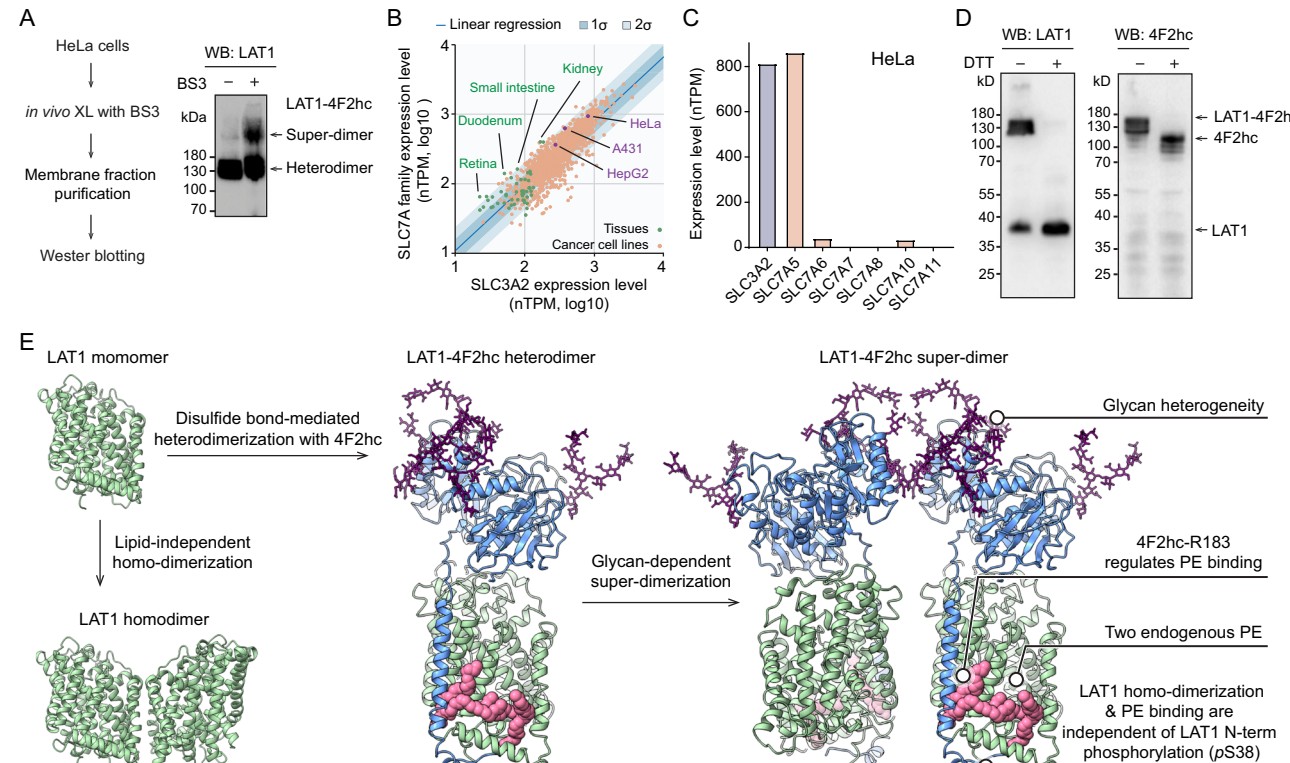

**Fig. 6 | Probing the endogenous LAT1-4F2hc assemblies. A** Western blotting analysis of the in vivo crosslinked (XL) endogenous LAT1-4F2hc in HeLa cells. An anti-LAT1 antibody was used to probe the endogenous LAT1-4F2hc assemblies without BS3 (-) and with BS3 (+) treatments. The experiment was repeated once. Source data are provided as a Source Data file. **B** Correlation of the mRNA expression levels of 4F2hc (SLC3A2) and its associated members of the SLC7A family (SLC7A5, SLC7A6, SLC7A7, SLC7A8, SLC7A9, SLC7A10) from human tissue and cell lines. The mRNA expression levels (measured in normalized transcripts per million, nTPM) of 4F2hc and SLC7As from 50 human tissues and 1205 cell lines were plotted as a scatter plot. The linear-fit trendline is plotted as a dark blue line. The shaded areas corresponding to one and two standard deviations ($\sigma$) of the linear regression are highlighted (dark blue and light blue, respectively). Source data are provided as a Source Data file. **C** Expression levels of SLC3A2 and SLC7A5/6/7/8/10/11 in HeLa cell lines. Source data are provided as a Source Data file. **D** Western blotting of the endogenous LAT1-4F2hc assemblies in HeLa cells without and with dithiothreitol (DTT) treatment. The experiments were repeated once. Source data are provided as a Source Data file. **E** Model of the dynamic assemblies of LAT1-4F2hc super-dimer with highlighted key findings. The LAT1 homo-dimer and LAT1-4F2hc super-dimer structures are proposed based on AlphaFold-Multimer and homology-modelling.

(Fig. 6B). Generally, 4F2hc is highly correlated with SLC7A transporter expression (Pearson $r = 0.8641$; $p < 0.0001$). Interestingly, however higher expression levels of SLC7A transporters are observed in many cell lines and tissues, particularly in the kidney, small intestine and retina implying that some free LAT1 may be present in these tissues.

To validate the presence of free endogenous LAT1 experimentally, we further assessed the ratios of endogenously expressed LAT1:LAT1-4F2hc in HeLa cells which have a slightly higher SLC7A5 mRNA level than the tissues considered above (Fig. 6C). We separated endogenous free LAT1 and the LAT1-4F2hc heterodimer with gel electrophoresis, without reducing the disulfide bond, and probed their presence with Western blotting (Fig. 6D). We observed both free LAT1 and heterodimeric LAT1-4F2hc, but no free 4F2hc subunits. We also confirmed the presence of both free LAT1 and LAT1-4F2hc heterodimers in two further cancer cell lines, HepG2 and A431 cells and again found no evidence for free 4F2hc subunits (Supplementary Fig. 17B–E).

Given the importance of LAT1-4F2hc in activating mTORC1 signalling upon translocation to the lysosome[35], we purified the lysosomal fraction from HeLa cells and investigated LAT1 assemblies in the lysosome (Supplementary Fig. 17F, G). Notably, the LAT1-4F2hc hetero-complex, but not free LAT1, is localized in the lysosome. Since interferon-γ (IFN-γ) is known to remodel lysosome activities and function[39], we stimulated and isolated lysosomes and tested again for free LAT1. We found that free LAT1, in the absence of additional reducing agents, is only present in the lysosome after stimulation with interferon-γ (IFN-γ). To explain this observation, we propose therefore that free LAT1 may be liberated by reduction from intact LAT1-4F2hc complexes by an IFN-γ-inducible lysosomal thiol reductase[40]. Taken together, these experiments imply that the assembly of LAT1-4F2hc is dynamic in vivo and that liberation of LAT1 is likely regulated by redox conditions in the lysosome.

## Discussion

Elucidating the complete assembly and modifications of membrane protein complexes is vital to understanding their function and regulation yet challenging to perform. Here, we developed a native MS approach to dissect the assembly of human LAT1-4F2hc, including defining heterogenous proteoforms, endogenous phospholipid binding sites and oligomerization status (Fig. 6E). We found that the LAT1-4F2hc complex carries various PTMs, namely N-glycosylation, phosphorylation and palmitoylation, as well as two PE lipid binding sites at the C-terminus of LAT1. Specifically, one of the PE lipids is at the heterodimer interface and interacts with 4F2hc-R183, is abolished by mutation, and likely promotes palmitoylation of LAT1-Cys187. Four highly branched N-glycans with 9 to 16 sialic acid residues contribute primarily to the heterogeneity of the LAT1-4F2hc complex and its super-dimerization propensity. Combining native MS, mass photometry and in silico modelling, we revealed that LAT1, similar to its prokaryotic homologue LeuT and Adic[17], is a homodimer. The attachment of 4F2hc to LAT1 perturbs the LAT1 dimerization interface, and induces N-glycan-mediated super-dimerization via the interfacial glycans in 4F2hc. Importantly, we showed how the terminal sialic acid

residues stabilize the super-dimer of LAT1-4F2hc. Our results further show that super-dimerization is enhanced at pH 5.0, which is similar to the acidic lysosome environment, implying that the LAT1-4F2hc super-dimer can survive in the lysosomal membrane where it can activate mTORC1 pathway[35]. Moreover, we revealed that the dynamic assembly of endogenous LAT1-4F2hc heterodimers and free LAT1 complexes in lysosome that is related to IFN-γ stimulation.

The modelled LAT1-4F2hc super-dimer structure suggests the interfacial glycan on Asn365 is related to dimer formation and shielded in the super-dimer interface. Interestingly, a recent study reported that the microheterogeneity of Asn365 is sensitive to metabolic influx, and related to the endocytosis processes[11]. Both results imply a connection between the nutrient influx and higher-order structure of LAT1-4F2hc via N-glycosylation. Similarly a previous study showed that N-glycosylation of rBAT-Asn575 is critical for b[0,+]AT-rBAT biogenesis[41]. We modelled possible N-glycan conformers on all five glycosylation sites on rBAT (Asn261, Asn322, Asn495, Asn513 and Asn575) (Supplementary Fig. 18). Only the N-glycan on Asn575 is at the super-dimer interface and extensively contacts the neighbouring rBAT subunit. Therefore, and by analogy, we propose that N-glycosylation also stabilizes and regulates the b[0,+]AT-rBAT super-dimerization which is essential for its maturation[18]. Interestingly, our Western blotting analysis of endogenous LAT1-4F2hc shows the presence of free LAT1. However, the free 4F2hc subunit is not proportional to the free LAT1 in HeLa and HepG2 cells (Fig. 6C and Supplementary Fig. 17C). This implies that the majority of endogenous 4F2hc protein is covalently linked to LAT1 and/or other solute carriers in vivo. We propose that disulfide bond formation may be critical for the assembly and/or subcellular localisation of LAT1-4F2hc complex in vivo. Further investigation into the glycosylation of LAT1 and rBAT in cancers and diseases will shed light on the regulation of heterodimeric amino acid transporters.

Given that the 4F2hc subunit tunes the transport specificity of LAT1[7], and the fact that many lysosomal proteins are highly N-glycosylated, or have highly N-glycosylated accessory subunits as in this case, 4F2hc presumably also exerts a protective function[42]. The observation of LAT1-4F2hc dynamic assemblies in lysosome further implies that the regulation afforded by 4F2hc becomes uncontrolled in cancer and inflammation when amino acid transport and reversible palmitoylation are known to be upregulated[43–45]. Considering the potential role of IFN-γ-inducible lysosomal thiol reductase in regulating free LAT1 in lysosome, further investigation of the effects of redox modulation on LAT1-4F2hc assemblies and subcellular localisation will provide a deeper understanding of amino acid influx for this important drug target.

From a methodological viewpoint glycosylation, which is prevalent in membrane transporters and receptors, typically impedes the study of endogenous membrane protein assemblies, particularly by native MS. The micro-heterogeneity (glycan composition, linkage and structure) and macro-heterogeneity (site occupancy) lead to overlapping peaks with different glycan composition and charge states. Therefore, previous studies have focused on analysing recombinantly expressed non-glycosylated or glycoengineered membrane proteins from insect cells/N-acetylglucosaminyltransferase 1 knock-out (GnT1[-/-]) cells[46–49]. Here, we extended high-resolution native MS to connect proteoforms and endogenous lipid binding to a highly glycosylated membrane protein with advanced data processing, gas-phase fragmentation, site-directed mutagenesis, and in silico modelling approaches. Importantly, for this LAT1-4F2hc complex from a human cell line, we resolved the heterogeneity of the glycan shield which, together with our observation of the assembly dynamics of endogenously expressed LAT1-4F2hc, allowed us to propose a protective mechanism for this shield in the lysosome. In so doing, we exemplify a native MS-based framework to decode mammalian multi-proteoform protein complexes and to relate heterogenous PTMs to protein assembly, lipid regulation and localisation.

## Methods

### Protein expression and purification

The LAT1-4F2hc complex expression and purification were adapted from the previous report[12]. The cDNAs of full-length LAT1 and 4F2hc were subcloned separately into pCAG with an N-terminal Flag tag fused to LAT1 and an N-terminal 8 × His tag fused to 4F2hc. HEK293F cells (Invitrogen) were cultured in SMM 293T-I medium (Sino Biological) at 37 °C under 5% $CO_2$ until the cell density reached $2.0 \times 10^6$ cells per ml. The LAT1 plasmid (0.75 mg) and 4F2hc plasmid (0.75 mg) were mixed with 3 mg of polyethylenimines (Polysciences) in 50 ml of fresh medium for 15–30 min and added to one liter of cell culture. The transfected cells were cultured for 48 h before collection. For purification of the LAT1–4F2hc complex, the cells were collected and lysed in the Tris buffer (25 mM Tris, 150 mM NaCl and pH 8.0) with aprotinin (0.8 μM, AMRESCO), pepstatin (2 μM, AMRESCO), and leupeptin (2.5 μM, AMRESCO). The membrane fraction was solubilized in 1% digitonin (Sigma) at 4 °C for 2 h and ultra-centrifuged at 100,000 g for 1 h to remove the insoluble debris. The supernatant was incubated with anti-Flag M2 affinity resin (Sigma, A2220) for LAT1-4F2hc affinity purification. The resin was then washed with wash buffer (25 mM Tris, 150 mM NaCl, 0.08% digitonin and pH 8.0). The protein complex was eluted with the wash buffer with 0.2 mg/ml Flag peptide. The eluent was then applied to nickel resin (Ni-NTA, Qiagen). The resin was washed with the wash buffer with 20 mM imidazole. The protein complex was then eluted from the nickel resin with the wash buffer with 300 mM imidazole. The eluent was then concentrated and subjected to SEC (Superose 6 Increase 10/300 GL, GE Healthcare) in the wash buffer. The peak fractions were pooled and concentrated for the following MS analysis.

### Native MS analysis

The LAT1-4F2hc protein was buffer-exchanged into 200 mM ammonium acetate (pH 7.0) with 2 mM OGNG using an Amicon 100 KD MWCO filter. The LAT1-4F2hc protein concentration was adjusted to about 4 μM for native MS analysis. The desalted protein complex (3 μl) was then loaded into a gold-coated borosilicate capillary prepared in-house and electrosprayed into a Q Exactive UHMR hybrid quadrupole-Orbitrap mass spectrometer (Thermo Fisher Scientific). The instrument settings were capillary voltage of 1.3 kV, UHV pressure of $5.5 \times 10^{-10}$ mbar, capillary temperature of 300 °C, and resolution of 17,500. The noise level was set at 3. Unless otherwise stated all the experiments were performed in the positive mode. Raw MS data were processed with Xcalibur 4.1.50 (Thermo Scientific). UniDec software v5.1.1[23] and iFAMs[24] were used for spectral deconvolution.

### Mass photometry analysis

Mass photometry experiments were performed using a Refeyn Two[MP] mass photometer (Refeyn Ltd)[50]. Data was acquired using AcquireMP (Refeyn Ltd, version 2.3). The instrument was calibrated using in-house prepared protein standards. The LAT1-4F2hc complex was buffer-exchanged to phosphate-buffered saline (PBS) with 36 μM GDN (2 × critical micelle concentration, CMC) and further diluted into PBS before the MP measurements. To identify the focal position, 18 μl PBS was added onto a clean microscope coverslip. Then, the protein sample (2 μl) was added and mixed with the 18 μl buffer on the coverslip. To ensure the low background, the final LAT1-4F2hc concentration for MP measurements was kept at -10 nM. The MP measurement was recorded for 60 sec. The raw data were processed using DiscoverMP (Refeyn Ltd, version 2.3) to integrate the binding events for mass determination and quantification.

## Reporting summary

Further information on research design is available in the Nature Portfolio Reporting Summary linked to this article.

## Data availability

The raw native MS data that generated in this study have been deposited in the Figshare database https://doi.org/10.6084/m9.figshare.24139461. The proteomics and lipidomics raw data that generated in this study have been deposited in the MassIVE database under the accession code MSV000094478. The atomic coordinates for structural illustration and simulation are available in the PDB database under accession code 6IRT (LAT1-4F2hc) and 6LI9 (b0,+AT-rBAT). Source data are provided with this paper.

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

## Acknowledgements
This research was funded in whole, or in part, by the Wellcome Trust grant no. 221795/Z/20/Z. For the purpose of Open Access, the author has applied a CC BY public copyright license to any Author Accepted Manuscript version arising from this submission. R.Y. is supported by the National Natural Science Foundation of China (82202517). J.S.P and A.S. would like to acknowledge the National Science Foundation (award CHE-1752994).

## Author contributions
D.W. and C.V.R. designed the research. D.W. performed MS experiments and data analysis. R.Y. and Y.L. performed protein purification. A.S. and J.S.P. performed data analysis. D.W. and S.S. performed Western blotting. C.V.R. and Q.Z. supervised the project. D.W. and C.V.R. wrote the manuscript with input from all authors.

## Competing interests
C.V.R. is a cofounder of and consultant at OMass Therapeutics. The remaining authors declare no competing interests.
