## [Peer Review File · Nature Communications]

Reviewers' Comments:

Reviewer #1:

Remarks to the Author:

Here, the authors use native mass spectrometry and mass photometry to study the LAT1-4F2hc complex. They identify and characterise the binding of endogenous lipids to the complex, and also report the presence of a 'super-dimer'. Intriguingly, after characterisation of the glycosylation of LAT1-4F2hc, they find that this 'super-dimerisation' is mediated by glycans, and is also pH dependent. Overall, I believe this work is both methodologically and biologically significant and interesting, but I do have a few suggestions for the authors to consider.

In the Introduction, the disulphide-linked LAT1-4F2hc complex is accurately referred to as a heterodimer. Likewise, in the paragraph on LAT1 dimerisation and PE binding, the LAT1 monomer and homodimer are mentioned. I thus found it a bit confusing that in the first couple of paragraphs of the Results, as well as the caption of Figure 1, the 'LAT1-4F2hc monomer' is mentioned, and I had to re-read a few sentences to realise that this 'monomer' actually referred to the heterodimer, and that this was being contrasted with the super-dimer. I would encourage the authors to go through the manuscript and clearly and consistently use the terms 'monomer', 'homodimer', 'heterodimer', and 'super-dimer' to avoid potential confusion.

Speaking of the super-dimer, a reader could be sceptical of its relevance if it were only supported by the native MS data, as no concentration is reported (or maybe I missed it) and many proteins can be made to stick together during electrospray ionisation. To be clear, I believe this super-dimer is real, given the consistent mass photometry data (acquired at 10 nM) and the literature report of something similar having been found for SLC7A9-SLC3A1, but reporting at least an estimated concentration of the protein in the native MS experiments would be useful.

The match between the observed and calculated mass differences in Figures 1E-F and Supplementary Figure 2 is quite impressive, given the mass of the heterodimer. However, in Figure 2D, the deviation from the expected mass differences for phosphorylation and addition of a GlcNAc are considerably larger. The complementary bottom-up data in Supplementary Figure 5 help, but can the authors comment on possible reasons for this greater mass error?

In the paragraph on 'Resolving endogenous lipid binding', it would be helpful to know what collision energy values were used, particularly to generate the spectra in Supplementary Figure 7A, main Figure 3C (assuming the complex doesn't fall apart under standard native MS conditions after disulphide reduction), and main Figures 4B, C, and F.

In this paragraph, the authors also write that 'the collisional activation is not sufficient to break the disulfide bond and disrupt LAT1-4F2hc complex'. I can clearly see the ejected LAT1 subunit in Supplementary Figure 7A though, so the complex is clearly disrupted to some extent. Whether this is merely stochastic, or indicative of different sub-populations with a different susceptibility to dissociating is an interesting question, but likely not easy to answer. This sentence should be rephrased to be a bit more nuanced though.

As a final point on the topic of Supplementary Figure 7A, it would be interesting to add a zoomed-in view, so that the reader can compare this to main Figure 3D.

Related to the above: Figure 4B is a spectrum of ejected LAT1 (much like Figure 3D), while Figures 4D and 4F show the heterodimer. It should be noted more explicitly in the text on page 6 whether or not increased collision energy was used to generate Figure 4B, or if the complex spontaneously falls apart under standard native MS conditions after disulphide reduction. If it does not, referring to the experiment simply as 'native MS analysis of the TCEP-treated mutant' on page 6 is potentially a bit confusing.

At the end of the Abstract, '(Word count 147)' presumably was meant to be deleted before submission.

Introduction, first sentence: 'Heteromeric amino acid transporters (HATs) are [...] comprised of a heavy [...] and a light subunit' – either 'are composed of' or 'comprise'.

In a few instances (e.g., last paragraph of the Introduction, caption of Figure 4) 'the C-terminal' is used as a noun. This should be replaced with (for example) 'the C-terminus' or 'the C-terminal region'.

On page 9, the authors write that the pH 5.0 conditions are not compatible with native MS. It is not immediately obvious why this is, so it would be good to clarify this.

In the paragraph on 'Probing endogenous LAT1-4F2hc assembly', the authors list 'their response to glycosylation, PTMs, palmitoylation and lipid binding'. Given that glycosylation and palmitoylation are examples of PTMs, there seems to be some redundancy in this sentence. Later in the same paragraph, 'free and complex forms of LAT1' sounds a bit strange to me.

In the discussion, there is a word missing in 'N-glycosylation, phosphorylation and palmitoylation, as well two PE lipid binding sites'.

In the Methods, 'using a Amicon 100 KD MWCO filter. The desalted protein complex (3 μ l) was then loaded into gold-coated borosilicate capillary' should be 'using an Amicon 100 KDa MWCO filter. The desalted protein complex (3 μ l) was then loaded into a gold-coated borosilicate capillary'.

For Supplementary Figure 3, can the authors provide scores (e.g., those reported by UniDec) for the deconvolution results? Also, Panel B of this figure might be more informative if the authors zoom in on a similar range to that shown in Panel F.

In Supplementary Figure 8B, '100 kD' should be '100 kDa'.

Reviewer #2:

Remarks to the Author:

The authors have impressively systematically probed the assembly of an important glycosylated membrane protein complex. The results link PTMs and lipid binding with regulation and trafficking of the LAT1-4F2hc super dimer. The authors were also able to model the MS-discovered PE lipids into previously obtained cryoEM density. They show free LAT1 present after IFN-gamma stimulation, consistent regulation by redox dynamics in vivo. A nice set of tools (MS, MP, computational) are used in a complementary fashion on this protein whose regulation is crucial for drug development and therapeutic delivery to tumors and the brain. Before this work, the superdimerization status and regulation through N-glycosylation, endogenous phospholipids, and PTMs were uncharacterized.

The work is noteworthy, significant both methodologically and biomedically, systematic, sound, and described in detail. Publication is recommended after minor revisions.

minor suggestions:

p. 3

1) describe digitonin and glyco-diosgenin with brief phrases so that a general reader does not have to look them up while reading

2) explain the delta of 73 (supplementary fig S2) more clearly

3) deconvolve instead of deconvolute

p. 4

4) explain for the general reader why the mass differences 74, 215 don't accurately mass expectations; same thing for p. 5 PE, PS, PI vs glycan proteoforms

p. 5

5) last sentence - clarify that these are not two different masses of phospholipids but actually correspond to peaks for one and two phospholipids

p. 6

6) p. 6, first paragraph, line 4, not fully clear what is meant by increased here and decreased in the fig caption; define prime vs no prime

p. 7

7) top of page, why the change to LMNG vs OGNG; did mutation necessitate the change?

p. 9

8) last paragraph, make hypothesis clearer above before you call it "this hypothesis"

general:

9) the authors are inconsistent in how they refer to LAT1-4F2hc, sometimes calling it a heterodimer, sometimes calling it a monomer, sometimes a heterodimeric monomer

10) Fig 5C, monome (typo) in desialylated

Reviewer #3:

Remarks to the Author:

Wu et al. have extensively worked in observing a complex that plays a significant role both physiologically and in some diseases such as cancer. The manuscript's detailed exploration, particularly using native mass spectrometry (MS) from the leading research lab, lends a robust and meticulous foundation to the study. The discovery of super-dimer formation (LAT1-4F2hc)₂ using Native MS is particularly noteworthy and presents a valuable addition to the field.

The manuscript also promises to advance future research with its detailed analysis of glycosylation patterns and lipid interactions, providing a foundation for further exploration into these critical biological processes. However, despite these strengths, the manuscript does not fully deliver on correlating the extensive data presented, thereby diminishing the clarity of the findings. The physiological relevance of the super-dimer formation remains unclear, and the existence of super-dimers within a physiological or cellular environment is not convincingly demonstrated.

Additionally, the analysis pertaining to the localization mechanism to lysosomes, while fascinating, suffers from a reliance on a limited range of experimental methods, which casts doubt on the reliability of these findings. The narrative of the manuscript also seems to force a volume of advanced MS data into a framework of biological relevance that is not sufficiently supported by the biochemical and cell biological methodologies employed. Overall, there is insufficient proof and discussion regarding the function and trafficking of the human LAT1-4F2hc complex.

With these points in mind, I suggest that the highly technical MS analysis data might find a better fit in a journal dedicated to such intricate details. The narrative concerning the physiological importance of the super-dimer would benefit from further experimental evidence, which could be addressed in a subsequent submission.

Pivotal Points:

1. There is an essential query regarding the existence of the super-dimer under physiological conditions. The observations reported appear to be limited to states where the complex is solubilized in detergents, specifically observed only in the presence of certain surfactants. It would be advantageous to demonstrate the presence of super-dimers without the aid of these solubilizing

agents to ascertain their physiological relevance.

2. The manuscript would benefit from a more robust discussion and demonstration of the super-dimer's function, trafficking, and its role across different cell types and tissues from a physiological standpoint.

Detailed Points:

1. In the introduction, the authors do not mention previous publications in which some of them were involved, which indicated that LAT1 alone does not possess transport function. It would be prudent to acknowledge and explain the absence of the function in the current context.

Interestingly, some studies have suggested that the 4F2hc subunit is not necessary for the transport or plasma membrane residence of LAT1 in vivo 3–5, raising the question of its functional relevance.

2. Reference 16 is utilized to discuss dimerization of the 4F2hc subunit; however, as it pertains only to the solubilized domain, it may not be the most appropriate reference for such a discussion.

3. The function of the complex in the presence of OGNG is not addressed. An analysis to confirm if there is, at least, substrate binding capability in this context would be valuable.

4. While the detailed analysis of the glycosylation of the LAT-4F2hc complex is commendable, it is suggested that the authors explore and explain variations in glycosylation patterns across different cell lines and tissues, as implied by the data.

5. Considering that multiple prior studies have shown that disulfide bond formation is not essential for the formation of the LAT-4F2hc complex, an analysis using variants employed in such studies might clarify the impact of lipids and glycosylation on complex formation.

6. It may be beneficial to experimentally demonstrate the effect of LAT1-C187 palmitoylation on transport function, especially in the context of 4F2hc-R183L mutations.

There is confusion regarding the role of 4F2hc-R183L in transport function, as indicated in previous studies by some authors.

These results lead us to propose that absence of the R183 lipid binding site, located close to the inner membrane bilayer, leads to palmitoylation to enable efficient trafficking of the complex to the plasma membrane 28.

7. The rationale for using LMNG with certain mutants, such as the 4M variant, is not sufficiently explained.

8. The lipid binding analysis is particularly intriguing. Considering previous studies emphasizing the importance of cholesterol for LAT1 function, it would be interesting to know if cholesterol was not observed in the study.

9. If the super-dimers observed via native MS are indeed lipid-free, one could question whether this represents an unnatural state devoid of function.

10. The claim that "free LAT1 is present" based solely on mRNA expression may be an overstatement and warrants a more cautious conclusion.

11. Figure 6: More detailed description of the experimental conditions of the western blot would be expected. For example, there is no mention of where DTT was added. Also, is there any DTT present in the adjacent lanes that could affect the results? Other than that, it is not proven that the disulfide bond was not cleaved in the process of manipulation.

12. A discussion on the intracellular localization of protein molecules would be more convincing with at least some microscopic observational evidence.
13. In Supplemental Figure 5, labeling for panels C and D appear to be missing.

> Reviewer #1 (Remarks to the Author):

1. Here, the authors use native mass spectrometry and mass photometry to study the LAT1-4F2hc complex. They identify and characterise the binding of endogenous lipids to the complex, and also report the presence of a 'super-dimer'. Intriguingly, after characterisation of the glycosylation of LAT1-4F2hc, they find that this 'super-dimerisation' is mediated by glycans, and is also pH dependent. Overall, I believe this work is both methodologically and biologically significant and interesting, but I do have a few suggestions for the authors to consider.

We thank the reviewer for the positive response.

2. In the Introduction, the disulphide-linked LAT1-4F2hc complex is accurately referred to as a heterodimer. Likewise, in the paragraph on LAT1 dimerisation and PE binding, the LAT1 monomer and homodimer are mentioned. I thus found it a bit confusing that in the first couple of paragraphs of the Results, as well as the caption of Figure 1, the 'LAT1-4F2hc monomer' is mentioned, and I had to re-read a few sentences to realise that this 'monomer' actually referred to the heterodimer, and that this was being contrasted with the super-dimer. I would encourage the authors to go through the manuscript and clearly and consistently use the terms 'monomer', 'homodimer', 'heterodimer', and 'super-dimer' to avoid potential confusion.

Thanks for the suggestion. We have rephrased 'LAT1-4F2hc monomer' to 'LAT1-4F2hc heterodimer' in the manuscript.

To clarify, we use 'monomer' for the LAT1 or 4F2hc subunits, 'homodimer' for the LAT1 dimer, 'heterodimer' for the LAT1-4F2hc complex, and 'super-dimer' for the LAT1-4F2hc super-dimer.

3. Speaking of the super-dimer, a reader could be sceptical of its relevance if it were only supported by the native MS data, as no concentration is reported (or maybe I missed it) and many proteins can be made to stick together during electrospray ionisation. To be clear, I believe this super-dimer is real, given the consistent mass photometry data (acquired at 10 nM) and the literature report of something similar having been found for SLC7A9-SLC3A1, but reporting at least an estimated concentration of the protein in the native MS experiments would be useful.

We have added protein concentration for native MS analysis in the Method part.

On page 13, "The LAT1-4F2hc protein concentration was adjusted to about 4 μ M for native MS analysis."

Furthermore, the presence of LAT1-4F2hc super-dimerization is also supported by the *in vivo* crosslinking (see point 28).

4. The match between the observed and calculated mass differences in Figures 1E-F and Supplementary Figure 2 is quite impressive, given the mass of the heterodimer. However, in Figure 2D, the deviation from the expected mass differences for phosphorylation and addition of a GlcNAc are

considerably larger. The complementary bottom-up data in Supplementary Figure 5 help, but can the authors comment on possible reasons for this greater mass error?

We have performed additional analysis to address this concern. We simulated the P4 proteoforms with phosphorylation and an additional GlcNAc residue, and P3 proteoforms with tri- and tetra-fucosylation. We further compared these four proteoforms with the LAT1-4F2hc native mass spectrum and included this result as the new Supplementary Figure 5D.

“Supplementary Figure 5. D) Native MS spectrum of the desialylated LAT1-4F2hc complex (5061-5073 m/z, charge state +27). The theoretical peaks of the P4 proteoforms (apo, phosphorylated and with additional GlcNAc) and P3 proteoforms (tri- and tetra-fucosylated) were simulated and plotted. The simulations show that the tri- and tetra-fucosylated P3 proteoforms overlap with the P4 proteoforms with phosphorylation and an additional GlcNAc, respectively.”

Notably, the tri- and tetra-fucosylated P3 proteoforms overlap with the P4 proteoforms with phosphorylation and an additional GlcNAc, respectively. We updated this result in the main text on page 5.

“The overlapped tri-fucosylated P3 proteoform (P3, aF3) and phosphorylated P4 proteoform (P4, aF0) result in a mass difference of 74.0 ± 0.7 Da (Figure 2D and Supplementary Figure 5D). Similarly, the mass difference of 214.7 ± 2.2 Da could be attributed to the overlapped tetra-fucosylated P3 proteoform (P3, aF4) and P4 proteoform (P4, aF0) with GlcNAc.”

5. In the paragraph on ‘Resolving endogenous lipid binding’, it would be helpful to know what collision energy values were used, particularly to generate the spectra in Supplementary Figure 7A, main Figure 3C (assuming the complex doesn’t fall apart under standard native MS conditions after disulphide reduction), and main Figures 4B, D, and F.

We thank the reviewer for this suggestion. Here, we have listed the collision energy values and capillary temperature for these spectra. We have also updated these parameters in the figures.

	Capillary Temp.	Source frag.	In-source Trapping	HCD
Suppl. Figure 8A	200 °C	200 V	50 V	150 V
Main Figure 3C	200 °C	150 V	75 V	0 V

Main Figure 4B	300 °C	50 V	0 V	200 V
Main Figure 4D	300 °C	30 V	0 V	100 V
Main Figure 4F	200 °C	150 V	0 V	0 V

For disulfide bond-reduced LAT1-4F2h, we observed the intact heterodimer, [LAT1-4F2hc] and super-dimer, [LAT1-4F2hc]₂ by mass photometry (as shown in Figure 3B). However, we didn't observe well-resolved LAT1-4F2hc complex after disulphide reduction by native MS. We have included an additional native mass spectrum of TCEP-treated LAT1-4F2hc under lower collisional energy (new Supplementary Figure 8D).

“Supplementary Figure 8. D) Native mass spectra of TCEP-reduced LAT1-4F2hc under different activation energies. We observed unresolved proteolipomicelles at lower activation energy (source fragmentation 50 V), and well-resolved LAT1 and 4F2hc subunits under higher activation energy (source fragmentation 150 V).”

6. In this paragraph, the authors also write that ‘the collisional activation is not sufficient to break the disulfide bond and disrupt LAT1-4F2hc complex’. I can clearly see the ejected LAT1 subunit in Supplementary Figure 7A though, so the complex is clearly disrupted to some extent. Whether this is merely stochastic, or indicative of different sub-populations with a different susceptibility to dissociating is an interesting question, but likely not easy to answer. This sentence should be rephrased to be a bit more nuanced though.

Thanks for the suggestion. Indeed, we observed dissociated LAT1 and 4F2hc subunits without TCEP-treatment. This implies a small population of recombinant LAT1-4F2hc complexes might not be covalently linked by the disulfide bond. We have rephrased the following sentences on page 6.

“The majority of LAT1-4F2hc heterodimers remain intact in the gas phase, however a small population of dissociated LAT1 and 4F2hc is observed following gas-phase activation and may be not covalently linked by the disulfide bond. However, the background is too high to annotate the adducts and proteoforms of each subunit, and the relatively high collision energy strips the bound lipid from the

LAT1 subunit (Supplementary Figure 8B). We, therefore treated the LAT1-4F2hc heterodimer with TCEP to fully reduce the disulfide bond in solution, prior to gas phase dissociation....”

7. As a final point on the topic of Supplementary Figure 7A, it would be interesting to add a zoomed-in view, so that the reader can compare this to main Figure 3D.

We have updated the new Supplementary Figure 8B. and compared the gas-phase dissociated LAT1 with the TCEP-reduced LAT1.

“Supplementary Figure 8B) Comparison of the native mass spectra of gas-phase dissociated LAT1 (top panel) and TCEP-reduced LAT1 (bottom panel). The higher collision energy used to dissociate LAT1 from the LAT1-4F2hc heterodimer also removes bound phospholipids.”

8. Related to the above: Figure 4B is a spectrum of ejected LAT1 (much like Figure 3D), while Figures 4D and 4F show the heterodimer. It should be noted more explicitly in the text on page 6 whether or not increased collision energy was used to generate Figure 4B, or if the complex spontaneously falls apart under standard native MS conditions after disulphide reduction. If it does not, referring to the experiment simply as ‘native MS analysis of the TCEP-treated mutant’ on page 6 is potentially a bit confusing.

We agree and have now labelled the collision energies used to obtain mass spectra in Figure 4B, 4D and 4F. We used a total of 250 V (source fragmentation 50 V, HCD 200 V) for Figure 4B, 130 V (source fragmentation 30 V, HCD 100 V) for Figure 4D and 150 V (source fragmentation 150 V) for Figure 4D. A significantly higher energy was used to dissociate LAT1 from the TCEP-treated LAT1-4F2hc-R183L mutant (Figure 4B).

The LAT1-4F2hc R183L mutant is also highly glycosylated. We are unable to directly distinguish lipid binding events on the intact heterodimeric mutant. Therefore, we treated the LAT1-4F2hc R183L mutant with TCEP and focused on probing the lipid binding to the LAT1 subunit (similar to Figure 3D). The 4M and 4M-ΔC mutants are not glycosylated. Therefore, we can directly analyze lipid binding to the heterodimer.

9. At the end of the Abstract, '(Word count 147)' presumably was meant to be deleted before submission.

We thank the reviewer for careful reading. We have deleted the word count from the Abstract part.

10. Introduction, first sentence: 'Heteromeric amino acid transporters (HATs) are [...] comprised of a heavy [...] and a light subunit' – either 'are composed of' or 'comprise'.

We have modified the corresponding sentence on page 2.

"Heteromeric amino acid transporters (HATs) are [...] composed of a heavy [...] and a light subunit..."

11. In a few instances (e.g., last paragraph of the Introduction, caption of Figure 4) 'the C-terminal' is used as a noun. This should be replaced with (for example) 'the C-terminus' or 'the C-terminal region'.

We have corrected the typo on page 3 and page 23.

12. On page 9, the authors write that the pH 5.0 conditions are not compatible with native MS. It is not immediately obvious why this is, so it would be good to clarify this.

We appreciate the reviewer's thoughtful consideration. Native MS can be applied to probe oligomerization of protein complexes in volatile buffers with different pH conditions (ref 36). However, it is challenging to (1) accurately quantify the partially resolved LAT1-4F2hc super-dimer using native MS and (2) directly relate the super-dimerization process to the pH of sample buffer. Therefore, we used the solution-based approach, mass photometry to accurately measure the pH-sensitive super-dimerization process of LAT1-4F2hc. We have added this explanation and a new reference (ref 36) on page 9.

"Therefore, we monitored super-dimerization of LAT1-4F2hc at pH 5.0 using MP in GDN micelles, as the partially resolved super-dimer could not be accurately quantified at acidic pH in native MS (ref 36)."

Ref 36. Gadzuk-Shea, M. M., Hubbard, E. E., Gozzo, T. A., & Bush, M. F. (2023). Sample pH Can Drift during Native Mass Spectrometry Experiments: Results from Ratiometric Fluorescence Imaging. *Journal of the American Society for Mass Spectrometry*, 34(8), 1675–1684.

13. In the paragraph on 'Probing endogenous LAT1-4F2hc assembly', the authors list 'their response to glycosylation, PTMs, palmitoylation and lipid binding'. Given that glycosylation and palmitoylation are examples of PTMs, there seems to be some redundancy in this sentence. Later in the same paragraph, 'free and complex forms of LAT1' sounds a bit strange to me.

We have modified the following two sentences.

Page 10. *"After elucidating the stoichiometry of LAT1-4F2hc assemblies in vitro, and their response to glycosylation, phosphorylation, palmitoylation and lipid binding..."*

Page 10. *"We also confirmed the presence of both free LAT1 and LAT1-4F2hc heterodimers in two further cancer cell lines, HepG2 and A431 cells"*

14. In the discussion, there is a word missing in 'N-glycosylation, phosphorylation and palmitoylation, as well two PE lipid binding sites'.

We have corrected the typo in this sentence.

Page 11 *"...namely N-glycosylation, phosphorylation and palmitoylation, as well as two PE lipid binding sites ..."*

15. In the Methods, 'using a Amicon 100 KD MWCO filter. The desalted protein complex (3 μ l) was then loaded into gold-coated borosilicate capillary' should be 'using an Amicon 100 KDa MWCO filter. The desalted protein complex (3 μ l) was then loaded into a gold-coated borosilicate capillary'.

We have now corrected the typo in this sentence.

Page 13 *"...with 2 mM OGNG using an Amicon 100 KD MWCO filter."*

16. For Supplementary Figure 3, can the authors provide scores (e.g., those reported by UniDec) for the deconvolution results? Also, Panel B of this figure might be more informative if the authors zoom in on a similar range to that shown in Panel F.

We have provided the UniScore (reported by UniDec) in the legend of Supplementary Figure 3B.

"...The major peaks are ~140 kDa. The UniScore (average peaks score) is 56.35 with R^2 of 0.99989."

We have also modified Supplementary Figure 3C showing a "zoom in" spectrum ranging from 136 to 150 kDa.

17. In Supplementary Figure 8B, '100 kD' should be '100 kDa'.

We have corrected the typo in the new Supplementary Figure 9B.

Reviewer #2 (Remarks to the Author):

The authors have impressively systematically probed the assembly of an important glycosylated membrane protein complex. The results link PTMs and lipid binding with regulation and trafficking of the LAT1-4F2hc super dimer. The authors were also able to model the MS-discovered PE lipids into previously obtained cryoEM density. They show free LAT1 present after IFN-gamma stimulation, consistent regulation by redox dynamics in vivo. A nice set of tools (MS, MP, computational) are used in a complementary fashion on this protein whose regulation is crucial for drug development and therapeutic delivery to tumors and the brain. Before this work, the superdimerization status and regulation through N-glycosylation, endogenous phospholipids, and PTMs were uncharacterized.

The work is noteworthy, significant both methodologically and biomedically, systematic, sound, and described in detail. Publication is recommended after minor revisions.

We thank the referee for the strong support to our work.

minor suggestions:

18. page3, describe digitonin and glyco-diosgenin with brief phrases so that a general reader does not have to look them up while reading

Thanks for the suggestion. We have added a brief introduction to those two detergents on page 3.

"...purified the protein complexes in a mild non-ionic detergent, digitonin, following an established protocol used for the Cryo-EM study of LAT1-4F2hc 12. We first recorded a mass spectrum on a Q Exactive-UHMR of the LAT1-4F2hc complex in glyco-diosgenin (GDN), a synthetic substitute for digitonin...."

19. explain the delta of 73 (supplementary fig S2) more clearly

We have modified Supplementary Figure 2A, 2B and added a more detailed legend to explain the simulation process and ~73 Da mass difference.

“Supplementary Figure 2. A) Simulation of LAT1-4F2hc proteoforms. The N-glycan structures, with different numbers of Fuc, Neu5Ac and Gal-GlcNAc units, are extracted from a previously published bottom-up MS analysis of LAT1-4F2hc (ref10), and used for simulation of LAT1-4F2hc proteoforms with four N-glycans at the intact protein level. The molecular weight of each proteoform is calculated based on the mass of protein backbone and combination of the four N-glycans. The simulated proteoforms are first sorted in ascending order, and then binned with a window of 5 Da. The mass difference between each proteoform (ΔM) is calculated and plotted as a scatter graph in panel B. The average mass difference between each proteoform is calculated as 72.95 ± 0.69 Da.”

20. deconvolve instead of deconvolute

We have corrected the typo on page 4 in the main text and on page 9, 10 in Supplementary Information.

21. Page 4, explain for the general reader why the mass differences 74, 215 don't accurately mass expectations; same thing for p. 5 PE, PS, PI vs glycan proteoforms

As discussed above (Point 3), we have added a new Supplementary Figure 5D to explain the mass differences of 74 and 215 (214.7 Da).

We have also performed further simulations to explain the mass differences between PE, PS and PI vs glycan proteoforms. We have updated this result in Supplementary Figure 7A.

“Supplementary Figure 7. A) Assignment of LAT1-4F2hc proteoforms and lipid adduct peaks. The isotopic envelopes of glycosylated LAT1-4F2hc, with different glycan structures and bound lipids, were simulated and plotted alongside the native mass spectrum of LAT1-4F2hc (charge state +27). Due to the inherent isotopic peaks of large protein complex and the limited resolution of the mass spectrum, LAT1-4F2hc heterodimer proteoforms and potential lipid-bound peaks overlap in the spectrum.”

22. Page 5, last sentence - clarify that these are not two different masses of phospholipids but actually correspond to peaks for one and two phospholipids

Thanks for the suggestion. We have modified the corresponding sentence on page 6.

“Importantly, we detected two phospholipid adduct peaks (730.1 ± 2.8 Da and 732.5 ± 0.5 Da) corresponding to two different phospholipids bound to the LAT1 monomer...”

23. Page 6, first paragraph, line 4, not fully clear what is meant by increased here and decreased in the fig caption; define prime vs no prime

We have modified the corresponding sentences on page 6

“The spectrum of dissociated 4F2hc subunits shows decreased abundances of the P5’ to P9’ proteoforms (Figure 3E, top panel), relative to the peaks assigned to the LAT1-4F2hc heterodimer (Figure 3E, bottom panel, P5 to P9 proteoform). This observation supports our conjecture that the lipid-bound species contribute to the increased abundances of the P5 to P9 series in the desialylated, intact LAT1-4F2hc heterodimer.”

We used ‘no prime’ species to describe major proteoforms of the LAT1-4F2hc heterodimer, and attributed the corresponding proteoform of the dissociated 4F2hc subunit (with the same monosaccharide compositions) to the prime species. We have updated this explanation in the Fig 3 legend.

“The corresponding proteoforms of the 4F2hc subunit (P1’ to P9’) and the LAT1-4F2hc heterodimer (P1 to P9) are aligned.”

24 page 7, top of page, why the change to LMNG vs OGNG; did mutation necessitate the change?

We found that the 4M and 4M- Δ C mutant heterodimer (lacking N-glycans) do not spray well in OGNG micelles. Therefore, we used LMNG for native MS analysis of these two mutants.

On page 7 *“Since the 4M mutant does not electrospray well in OGNG, we analysed the mutant in lauryl maltose neopentyl glycol (LMNG) detergent...”*

25. page 9, last paragraph, make hypothesis clearer above before you call it "this hypothesis"

We have modified the corresponding sentence on page 10.

“To validate the presence of the free endogenous LAT1 subunit experimentally, we further assessed the ratios of endogenously expressed LAT1:LAT1-4F2hc in HeLa cells...”

general:

26. the authors are inconsistent in how they refer to LAT1-4F2hc, sometimes calling it a heterodimer, sometimes calling it a monomer, sometimes a heterodimeric monomer

Thanks for the suggestion. We have modified the manuscript and defined the use of ‘monomer’ for LAT1 subunit or 4F2hc subunit, ‘homodimer’ for LAT1 dimer, ‘heterodimer’ for LAT1-4F2hc complex, and ‘super-dimer’ for LAT1-4F2hc super-dimer.

27. Fig 5C, monome (typo) in desialylated

We have modified Figure 5C and corrected the typo.

Reviewer #3 (Remarks to the Author):

Wu et al. have extensively worked in observing a complex that plays a significant role both physiologically and in some diseases such as cancer. The manuscript's detailed exploration, particularly using native mass spectrometry (MS) from the leading research lab, lends a robust and meticulous foundation to the study. The discovery of super-dimer formation (LAT1-4F2hc)₂ using Native MS is particularly noteworthy and presents a valuable addition to the field.

The manuscript also promises to advance future research with its detailed analysis of glycosylation patterns and lipid interactions, providing a foundation for further exploration into these critical biological processes. However, despite these strengths, the manuscript does not fully deliver on correlating the extensive data presented, thereby diminishing the clarity of the findings. The physiological relevance of the super-dimer formation remains unclear, and the existence of super-dimers within a physiological or cellular environment is not convincingly demonstrated.

We thank the reviewer for highlighting our main findings and providing constructive suggestions. We have now addressed all concerns point-by-point. We have performed lipidomics analysis and found cholesterol is also co-purified with LAT1-4F2hc, in line with previous Cryo-EM observations. Moreover, we present further evidence for the endogenous LAT1-4F2hc super-dimer in HeLa cells using *in vivo* crosslinking, Western blotting and proteomics approaches. Details are provided in response to the points below.

Additionally, the analysis pertaining to the localization mechanism to lysosomes, while fascinating, suffers from a reliance on a limited range of experimental methods, which casts doubt on the reliability of these findings. The narrative of the manuscript also seems to force a volume of advanced MS data into a framework of biological relevance that is not sufficiently supported by the biochemical and cell biological methodologies employed. Overall, there is insufficient proof and discussion regarding the function and trafficking of the human LAT1-4F2hc complex. With these points in mind, I suggest that the highly technical MS analysis data might find a better fit in a journal dedicated to such intricate details. The narrative concerning the physiological importance of the super-dimer would benefit from further experimental evidence, which could be addressed in a subsequent submission.

We appreciate the review's expertise and suggestions to further define the physiological relevance of the LAT1-4F2hc super-dimer. The LAT1-4F2hc heterodimer was first described in the late 90s (Mastroberardino et al. & Kanai et al. cited below). Due to the lack of suitable methods for *in vitro* and *in vivo* measurements in the last two decades, LAT1-4F2hc was considered a heterodimer without further higher-order structures. By contrast, the homologous b⁰⁺AT-rBAT complex has been shown to form a super-dimer.

The reviewer comments that we force a volume of advanced MS data into a framework of biological relevance. This is an interesting point but it is important to point out that such MS data would not have been possible until very recent advances in high-resolution native MS. Now for the first time, we are able to directly probe the proteoforms and endogenous ligand binding on intact membrane protein complexes and relate specific proteoform and/or lipid bindings to the regulation of protein assembly. Together with our novel data processing procedures, we are able to elucidate the complete repertoire of PTMs and endogenous lipids within LAT1-4F2hc assemblies. Importantly, we revealed lipid-

independent LAT1 homo-dimerization and N-glycan-mediated LAT1-4F2hc super-dimerization. While we have strengthened our biochemical data to also demonstrate the presence of the LAT1-4F2hc super-dimer on cell membranes. Further advances in biophysical and biochemical methodologies are needed to enable the direct observation of glycan-dependent super-dimer *in vivo*. Our native MS data motivates further experiments. However, these studies are out of the scope of this manuscript which is already substantial.

Mastroberardino, L., Spindler, B., Pfeiffer, R., Skelly, P. J., Loffing, J., Shoemaker, C. B., & Verrey, F. (1998). Amino-acid transport by heterodimers of 4F2hc/CD98 and members of a permease family. *Nature*, 395(6699), 288–291.

Kanai, Y., Segawa, H., Miyamoto, K. I., Uchino, H., Takeda, E., & Endou, H. (1998). Expression cloning and characterization of a transporter for large neutral amino acids activated by the heavy chain of 4F2 antigen (CD98). *The Journal of Biological Chemistry*, 273(37), 23629–23632.

We have also included this explanation in the main text on page 2.

“...The super-dimerization status of the LAT1-4F2hc heterodimer and its regulation through its heterogeneous N-glycosylation repertoire, endogenous phospholipids and PTM status, however, remain challenging to study via established structural biology approaches To overcome this complexity, here we develop and apply a high-resolution native MS approach to probe the assembly of the LAT1-4F2hc complex ...”

> Pivotal Points:

28. There is an essential query regarding the existence of the super-dimer under physiological conditions. The observations reported appear to be limited to states where the complex is solubilized in detergents, specifically observed only in the presence of certain surfactants. It would be advantageous to demonstrate the presence of super-dimers without the aid of these solubilizing agents to ascertain their physiological relevance.

Thanks for the comments which are well made. To further support our findings, we have now performed an *in vivo* cross-linking experiment to probe endogenous LAT1-4F2hc super-dimerization on HeLa cell membranes. Indeed, using Western blotting and affinity-purification MS analysis, after *in vivo* cross-linking, we were able to observe endogenous LAT1-4F2hc super-dimer formation on HeLa cell membranes. We included this important result in the main text on page 10 and in Supplementary Figure 16.

Page 10 *“Firstly, we validated super-dimerization of endogenous LAT1-4F2hc in HeLa cells. We incubated the cells with BS³ (bissulfosuccinimidyl suberate) to crosslink cell surface LAT1-4F2hc complexes and probed their super-dimerization status with Western blotting (Supplementary Figure 16A). We found that LAT1-4F2hc forms a large complex above 200 kDa on HeLa cell membrane (Supplementary Figure 16B). We further affinity-purified crosslinked LAT1-4F2hc complexes using anti-LAT1 antibody and analysed the band above 200 kDa (Supplementary Figure 16C). We only identified LAT1 and 4F2hc subunits as the most abundant proteins (besides IgG) in this band. This result implies*

that the large crosslinked complex is the LAT1-4F2hc super-dimer (Supplementary Figure 16D). This result supports that LAT1-4F2hc forms super-dimer *in vivo*.”

“Supplementary Figure 16. Probing endogenous LAT1-4F2hc super-dimerization on cell membrane. A) Flowchart of *in vivo* crosslinking (XL) and Western blotting (WB). The membrane-insoluble BS³ reagent can only crosslink cell surface proteins. B) Western blotting analysis using an anti-LAT1 antibody of LAT1-4F2hc assemblies, control without BS³ and with BS³. C) Flowchart of affinity purification-MS of *in vivo* crosslinked LAT1-4F2hc complexes. D) Top-five identified proteins based on their iBAQ (intensity-based absolute quantification) value using proteomics analysis. The large hydrophobic transmembrane domains of LAT1 could not be accessed by proteomics analysis with trypsin digestion. Therefore, the iBAQ value of LAT1 evaluated by proteomics is significantly lower than that of 4F2hc.”

29. The manuscript would benefit from a more robust discussion and demonstration of the super-dimer’s function, trafficking, and its role across different cell types and tissues from a physiological standpoint.

As discussed above, the study of super-dimer function and trafficking in different cell types and tissues is beyond the scope of this study and would be speculative at this stage.

We have rephrased the title to “The complete assembly of the human LAT1-4F2hc complex provides insights into its regulation, function and localisation”.

> Detailed Points:

30. In the introduction, the authors do not mention previous publications in which some of them were involved, which indicated that LAT1 alone does not possess transport function. It would be prudent to acknowledge and explain the absence of the function in the current context. Interestingly, some studies have suggested that the 4F2hc subunit is not necessary for the transport or plasma membrane residence of LAT1 in vivo 3–5, raising the question of its functional relevance.

We appreciate the reviewer's suggestion. We have rephrased the corresponding sentences and added relevant references on page 2.

“Interestingly, the regulatory roles of 4F2hc on the transport activity, selectivity and plasma membrane residence of LAT1 are not fully understood, given the conflicting findings obtained from in vitro and in vivo studies (ref 3–7).”

Ref 3. Napolitano, L. et al. LAT1 is the transport competent unit of the LAT1/CD98 heterodimeric amino acid transporter. *Int. J. Biochem. Cell Biol.* 67, 25–33 (2015).

Ref 4. Campbell, W. A. & Thompson, N. L. Overexpression of LAT1/CD98 light chain is sufficient to increase system L-amino acid transport activity in mouse hepatocytes but not fibroblasts. *J. Biol. Chem.* 276, 16877–84 (2001).

Ref 5. Boado, R. J. et al. Site-directed mutagenesis of cysteine residues of large neutral amino acid transporter LAT1. *Biochim. Biophys. Acta* 1715, 104–10 (2005).

Ref 6. Pfeiffer, R. et al. Functional heterodimeric amino acid transporters lacking cysteine residues involved in disulfide bond. *FEBS Lett.* 439, 157–62 (1998).

Ref 7. Kantipudi, S., Jeckelmann, J.-M., Ucurum, Z., Bosshart, P. D. & Fotiadis, D. The Heavy Chain 4F2hc Modulates the Substrate Affinity and Specificity of the Light Chains LAT1 and LAT2. *Int. J. Mol. Sci.* 21, 7573 (2020).

31. Reference 16 is utilized to discuss dimerization of the 4F2hc subunit; however, as it pertains only to the solubilized domain, it may not be the most appropriate reference for such a discussion.

Thanks for the suggestion. We have rephrased the sentences on page 2, cited two further references and discussed the rBAT-mediated super-dimerization of heteromeric amino acid transporters (HATs).

“Moreover LAT1, as one of the LeuT-fold amino acid transporters, is proposed to be assembled as a dimer in the membrane (ref 17), while the rBAT (SLC3A1), a 4F2hc homology heavy chain, has also been shown to mediate super-dimerization of HAT heterodimers (ref 15,18).”

Ref 15

*“Yan, R. et al. Cryo-EM structure of the human heteromeric amino acid transporter b^{0,+}AT-rBAT. *Sci. Adv.* 6, 1–11 (2020).”*

Ref 18

*“Lee, Y. et al. Ca²⁺-mediated higher-order assembly of heterodimers in amino acid transport system b^{0,+} biogenesis and cystinuria. *Nat. Commun.* 13, 2708 (2022).”*

32. The function of the complex in the presence of OGNG is not addressed. An analysis to confirm if there is, at least, substrate binding capability in this context would be valuable.

We used OGNG in native MS analysis to minimize the detergent interference of the heterogeneous proteoforms of the LAT1-4F2hc heterodimer, allowing us to resolve detailed PTM information, two PE lipid binding events and super-dimer formation. The PTM status of LAT1-4F2hc is not influenced by the presence of detergent. Furthermore, we validated the PE binding to the non-glycosylated mutants with LMNG, which is commonly used for purifying and structural analysis of heteromeric amino acid transporters (ref 12, 14, 16, 18). We also confirmed the LAT1-4F2hc super-dimer in solution using mass photometry with GDN, another widely used detergent for Cryo-EM study of LAT1-4F2hc, and on cell membranes using *in vivo* crosslinking. Therefore, these results suggest OGNG does not impact key lipid binding events and super-dimerization of LAT1-4F2hc complex.

The substrate binding properties and transport functions of LAT1-4F2hc in detergent micelles are beyond the scope of this manuscript.

33. While the detailed analysis of the glycosylation of the LAT-4F2hc complex is commendable, it is suggested that the authors explore and explain variations in glycosylation patterns across different cell lines and tissues, as implied by the data.

We analysed the glycosylation status of recombinant LAT1-4F2hc complex and correlated the glycan heterogeneity with LAT1-4F2hc super-dimerization. Further investigation of glycosylation patterns across different cell lines and tissues will undoubtedly inform LAT1-4F2hc assembly and regulation. We have added a discussion on page 12.

“Further investigation into the glycosylation of LAT1 and rBAT in cancers and diseases will shed light on the regulation of heterodimeric amino acid transporters.”

34. Considering that multiple prior studies have shown that disulfide bond formation is not essential for the formation of the LAT-4F2hc complex, an analysis using variants employed in such studies might clarify the impact of lipids and glycosylation on complex formation.

We appreciate the reviewer for this valuable point. We have added a discussion on page 11.

“Interestingly, our Western blotting analysis of endogenous LAT1-4F2hc shows the presence of free LAT1. However, the free 4F2hc subunit is not proportional to the free LAT1 in HeLa and HepG2 cells (Figure 6C and Supplementary Figure 17C). This implies that the majority of endogenous 4F2hc protein is covalently linked to LAT1 and/or other solute carriers in vivo. We propose that disulfide bond formation may be critical for the assembly and/or subcellular localisation of LAT1-4F2hc complex in vivo.”

35. It may be beneficial to experimentally demonstrate the effect of LAT1-C187 palmitoylation on transport function, especially in the context of 4F2hc-R183L mutations.

Thanks for the suggestion. The stoichiometry of the palmitoylated form is relatively low, as shown in the main Figure 4B (less than 20%). However, there is no established method to purify/enrich the palmitoylated form for a functional assay. Our previous paper (ref 12) demonstrated that the 4F2hc-R183L mutation, which leads to palmitoylation, impairs LAT1-4F2hc transporter activity.

36. There is confusion regarding the role of 4F2hc-R183L in transport function, as indicated in previous studies by some authors. These results lead us to propose that absence of the R183 lipid binding site, located close to the inner membrane bilayer, leads to palmitoylation to enable efficient trafficking of the complex to the plasma membrane 28.

We reported that 4F2hc-R183L impairs the transport activity of LAT1-4F2hc complex (ref 12). We rephrase the corresponding sentences on page 7.

“These results lead us to propose that absence of the R183 lipid binding site, located close to the inner membrane bilayer, leads to palmitoylation suggesting that this region may play a role in efficient trafficking of the complex to the plasma membrane...”

37. The rationale for using LMNG with certain mutants, such as the 4M variant, is not sufficiently explained.

The non-glycosylated 4M and 4M-ΔC variants do not electrospray well in ONGNG detergents, possibly due to the lack of N-glycans. Therefore, we analysed these two variants in LMNG. We have added the explanation on page 7.

“Since the 4M mutant does not electrospray well in OGNG, we analysed the mutant in lauryl maltose neopentyl glycol (LMNG) detergent...”

38. The lipid binding analysis is particularly intriguing. Considering previous studies emphasizing the importance of cholesterol for LAT1 function, it would be interesting to know if cholesterol was not observed in the study.

Thanks for this constructive suggestion. We performed additional lipidomics analysis and observed that cholesterol is also co-purified with LAT1-4F2hc complex. We have updated this finding in Supplementary Figure 7, and described the result on page 5.

“Moreover, we also identified cholesterol in co-purified lipids (Supplementary Figure 7B to D), in line with the previous observation (ref 12,13).”

“Supplementary Figure 7. B) Full mass spectrum of cholesterol molecule ([M-H₂O+H]⁺, 369.3518 m/z) identified in LAT1-4F2hc sample. C) Comparison of the extracted ion chromatograms of 369.3521 m/z in LAT1-4F2hc sample and cholesterol standard (on-column amount of 3 ng). D) The comparison of tandem mass spectra of 369.35 m/z in LAT1-4F2hc sample and cholesterol standard. These suggest cholesterol as a co-purified lipid of LAT1-4F2hc complex.”

39. If the super-dimers observed via native MS are indeed lipid-free, one could question whether this represents an unnatural state devoid of function.

We apologise for this misunderstanding. The super-dimer is not “lipid-free”. We observed two lipids binding to the LAT1-4F2hc super-dimer, and attributed these two lipids to PE, the same lipids observed in LAT1-4F2hc heterodimer (Supplementary Figure 13). We modified the corresponding sentences on page 8.

“This result is further supported by the observation that the LAT1-4F2hc super-dimer is free of interfacial lipid when observed in native MS (Supplementary Figure 13).”

We have also iterated this on page 4.

“...we can use the LAT1-4F2hc proteoforms to simulate the super-dimer heterogeneity using a binomial model 25 (Supplementary Figure 4). Using this approach, of matching simulated and raw data, we found that the theoretical average mass of the LAT1-4F2hc super-dimer (calculated from two times the measured mass of the heterodimer (280954 Da) agrees well with the MS-measured value (280934 ± 109 Da). Our mass measurements, therefore imply that the super-dimer does not require additional interfacial lipids or ligands...”

40. The claim that "free LAT1 is present" based solely on mRNA expression may be an overstatement and warrants a more cautious conclusion.

We thank the reviewer for this suggestion. While the dataset for absolute quantification of LAT1 and 4F2hc protein expression levels across cancer cell lines and tissues is not available, we have compared the relative protein expression and mRNA levels of LAT1 and 4F2hc across 375 cancer cell lines (new

Supplementary Figure 17A). We observed a positive correlation between LAT1 and 4F2hc protein expression levels and their respective mRNA levels. This suggests that we may estimate the LAT1 and 4F2hc protein expression levels based on their mRNA expression levels.

We modified the corresponding sentence in the main text on page 10.

"...some free LAT1 may be present in these tissues."

"Supplementary Figure 17. A) Correlation of the mRNA and protein expression levels of LAT1 (SLC7A5) and 4F2hc (SLC3A2) in 375 cancer cell lines. The relative protein expression levels (measured in TMT ratio) and mRNA levels (measured in normalized transcripts per million, nTPM) of LAT1 and 4F2hc were extracted from the Cancer Cell Line Encyclopedia (ref 16) and plotted as a scatter plot. The linear-fit trendline is highlighted (dark blue). The shaded area indicates the 95% confidence interval."

Ref 16. Nusinow, D. P. et al. Quantitative Proteomics of the Cancer Cell Line Encyclopedia. Cell 180, 387-402.e16 (2020).

41. Figure 6: More detailed description of the experimental conditions of the western blot would be expected. For example, there is no mention of where DTT was added. Also, is there any DTT present in the adjacent lanes that could affect the results? Other than that, it is not proven that the disulfide bond was not cleaved in the process of manipulation.

We added incubated solubilized membrane proteins with DTT just before the SDS-PAGE experiment. We modified the corresponding sentences in the Supplementary Methods.

"The membranes of A431, HeLa and HepG2 cells were incubated with 1× NuPAGE LDS sample buffer (Pierce) for 30 min at room temperature and centrifuged at 12000 g for 5 min to remove the undissolved particles. To reduce the disulfide bond in LAT1-4F2hc complex, the samples in LDS sample buffer were further incubated with 10 mM DTT at 56 °C for 10 min. The membrane proteins were then separated by NuPAGE 4 to 12%..."

The presence of DTT does not affect the integrity of the LAT1-4F2hc complex in the adjacent lane, as shown in Figure 6D (control), Supplementary Figure 17C (HepG2 lysate and lysosome) and Supplementary Figure 17E (A431 lysosome).

42. A discussion on the intracellular localization of protein molecules would be more convincing with at least some microscopic observational evidence.

Thanks for the suggestion. We agree that microscopic observation can provide direct evidence of protein subcellular localisation. However, the microscopic approaches to studying endogenous protein assemblies are limited (Nature Reviews Methods Primers, 2021). We didn't find an antibody that only recognizes the LAT1-4F2hc heterodimer but not the super-dimer. Moreover, the antibody that only immunolabels free LAT1 but not LAT1-4F2hc heterodimer is not available. Therefore, we have elected to use Western blotting to study the dynamic assembly of endogenous LAT1-4F2hc. We also cited a previous publication studying the lysosomal localisation of over-expressed recombinant LAT1-4F2hc via confocal microscopy (ref 35).

Lelek, M., Gyparaki, M. T., Beliu, G., Schueder, F., Griffié, J., Manley, S., Jungmann, R., Sauer, M., Lakadamyali, M., & Zimmer, C. (2021). Single-molecule localization microscopy. *Nature Reviews Methods Primers*, 1(1).

43. In Supplemental Figure 5, labeling for panels C and D appear to be missing.

We have added the labelling for panels C and D in Supplementary Figure 6.

Reviewers' Comments:

Reviewer #1:

Remarks to the Author:

In this revised version, the authors have adequately addressed all of my comments. The additional analysis and experiments performed in response to comments by myself and the other reviewers have significantly strengthened this interesting manuscript in my opinion.

I have one very minor suggestion regarding the text on line 101: 'the origin of these unresolved proteoforms' seems to refer to the 'resolved proteoforms' on line 99; therefore, rephrasing this as 'unidentified proteoforms' or maybe just 'proteoforms' on line 101 might further improve the clarity of the sentence.

Reviewer #2:

Remarks to the Author:

I was already impressed by this manuscript in its original form. The authors have prepared an excellent revision that addresses the reviewers' comments well. Importantly, the authors added new text and text clarifications, additional references, data, data analysis, and several new supplementary figures (in vivo crosslinking, new TCEP data, mass photometry data, additional Western blot analysis, additional lipid analysis to show cholesterol co-purifies). Some items requested by reviewers were deemed out of scope and I agree. The manuscript is even stronger than before and is suitable for publication.

Reviewer #3:

Remarks to the Author:

The authors have responded to the reviewers' comments with great care and consideration, which is commendable. Identifying the LAT1-4F2hc super-dimer, previously observed as an unidentified band in Western blots in past studies, using state-of-the-art techniques, is particularly praiseworthy. Clarification of this point makes the section of this manuscript on MS worthy of publication.

However, there remain issues concerning the physiological and biochemical analyses presented. Especially, the emphasis on the effects of interferon-gamma, based on a very limited set of experiments, is a point of concern. Despite the disparity in the level of analysis with MS, one might question if the significance attributed to these effects in the Abstract and Discussion sections might be overstated. Furthermore, the authors seem to have a misunderstanding about microscopy techniques, and their refusal to accept the reviewers' suggestions on this matter is problematic for improving the reliability of the description of the superdimer's physiology. The manuscript would be more coherent and scientific if the story on the effects of interferon-gamma were removed.

Regarding the additional cross-linking experiments conducted, while not entirely sufficient, they merit some recognition. However, the BS3 linker the authors used is relatively long, at approximately 11 angstroms. Typically, cross-linkers of this length do not indicate direct interaction, raising the question of whether this length is consistent with the model created by the authors using Alpha fold.

Reviewer #1 (Remarks to the Author):

In this revised version, the authors have adequately addressed all of my comments. The additional analysis and experiments performed in response to comments by myself and the other reviewers have significantly strengthened this interesting manuscript in my opinion.

I have one very minor suggestion regarding the text on line 101: 'the origin of these unresolved proteoforms' seems to refer to the 'resolved proteoforms' on line 99; therefore, rephrasing this as 'unidentified proteoforms' or maybe just 'proteoforms' on line 101 might further improve the clarity of the sentence.

We thank the reviewer for the careful reading. We have rephrased 'the origin of these unresolved proteoforms' to 'unidentified proteoforms' on page 3.

Reviewer #2 (Remarks to the Author):

I was already impressed by this manuscript in its original form. The authors have prepared an excellent revision that addresses the reviewers' comments well. Importantly, the authors added new text and text clarifications, additional references, data, data analysis, and several new supplementary figures (in vivo crosslinking, new TCEP data, mass photometry data, additional Western blot analysis, additional lipid analysis to show cholesterol co-purifies). Some items requested by reviewers were deemed out of scope and I agree. The manuscript is even stronger than before and is suitable for publication.

We appreciate the reviewer's recommendation of our manuscript and the recognition that some requests by reviewers are beyond the scope of this manuscript.

Reviewer #3 (Remarks to the Author):

The authors have responded to the reviewers' comments with great care and consideration, which is commendable. Identifying the LAT1-4F2hc super-dimer, previously observed as an unidentified band in Western blots in past studies, using state-of-the-art techniques, is particularly praiseworthy. Clarification of this point makes the section of this manuscript on MS worthy of publication.

We thank the reviewer for the positive comments on the *in vivo* crosslinking data and our revised manuscript. Hence, we moved the WB of *in vivo* crosslinking data (previous Supplementary Figure 16A and B) to Figure 6A.

However, there remain issues concerning the physiological and biochemical analyses presented. Especially, the emphasis on the effects of interferon-gamma, based on a very limited set of experiments, is a point of concern. Despite the disparity in the level of analysis with MS, one might question if the significance attributed to these effects in the Abstract and Discussion sections might be overstated. Furthermore, the authors seem to have a misunderstanding about microscopy techniques, and their refusal to accept the reviewers' suggestions on this matter is problematic for improving the reliability of the description of the superdimer's physiology. The manuscript would be more coherent and scientific if the story on the effects of interferon-gamma were removed.

We moved the Western blotting of IFN- γ stimulated lysosome (Figure 6c, left panel) to Supplementary Figure 17G. We rephrased the sentences about the effects of IFN- γ in Abstract and Discussion parts.

On page 1, Abstract section,

'Combining native MS with mass photometry (MP), we reveal that super-dimerization is sensitive to pH, and modulated by complex N-glycans on the 4F2hc subunit. We further validate the dynamic assemblies of LAT1-4F2hc on plasma membrane and in the lysosome.'

Results section,

On page 10, 'Notably, the LAT1-4F2hc hetero-complex, but not free LAT1, is localized in the lysosome.'

On page 11, 'To explain this observation, we propose therefore that free LAT1 may be liberated by reduction from intact LAT1-4F2hc complexes by an IFN- γ -inducible lysosomal thiol reductase. Taken together, these experiments imply that the assembly of LAT1-4F2hc is dynamic in vivo and that liberation of LAT1 is likely regulated by redox conditions in the lysosome.'

On page 11, 'Moreover, we revealed that the dynamic assembly of endogenous LAT1-4F2hc heterodimers and free LAT1 complexes in lysosomes that is related to IFN- γ stimulation.'

On page 12, 'The observation of LAT1-4F2hc dynamic assemblies in lysosome further implies that the regulation afforded by 4F2hc becomes uncontrolled in cancer and inflammation when amino acid transport and reversible palmitoylation are known to be upregulated. Considering the potential role of IFN- γ -inducible lysosomal thiol reductase in regulating free LAT1 in lysosome, further investigation of the effects of redox modulation on LAT1-4F2hc assemblies and subcellular localisation will provide a deeper understanding of amino acid influx for this important drug target.'

Regarding the additional cross-linking experiments conducted, while not entirely sufficient, they merit some recognition. However, the BS3 linker the authors used is relatively long, at approximately 11 angstroms. Typically, cross-linkers of this length do not indicate direct interaction, raising the question of whether this length is consistent with the model created by the authors using Alpha fold.

We have already provided the evidences of LAT1-4F2hc super-dimerization using different approaches, including native MS, mass photometry, *in silico* modelling and *in vivo* crosslinking.

The BS3 crosslinker used in this study carries a 11 Å linker arm. It is one of the most commonly used crosslinker in structural MS studies. Considering the dynamics of lysine side chain and protein conformation flexibility, a distance constraint of 26-30 Å between C α atoms is considered as appropriate (Merkley, E. D. et al). Therefore, we listed the lysine residues (C α) between two LAT1-4F2hc heterodimers within 28 Å for the super-dimer models (Supplementary Figure 14B and 14D). Notably, this data show that all four super-dimer models can be crosslinked by BS3 with 11 Å spacer arm.

	Lysine 1	Lysine 2	C α -C α distance (Å)
Homologous model	LAT1-Lys191	LAT1-Lys191	13
	LAT1-Lys191	LAT1-Lys204	27
AlphaFold model 1	LAT1-Lys191	4F2hc-Lys171	27
	LAT1-Lys191	LAT1-Lys191	27
	LAT1-Lys191	LAT1-Lys498	27
AlphaFold model 2	4F2hc-Lys396	4F2hc-Lys592	28
AlphaFold model 3	LAT1-Lys191	LAT1-Lys191	17
	4F2hc-Lys396	4F2hc-Lys396	21
	4F2hc-Lys396	4F2hc-Lys368	27

Reference:

Merkley, E. D. et al. Distance restraints from crosslinking mass spectrometry: Mining a molecular dynamics simulation database to evaluate lysine-lysine distances. *Protein Sci.* 23, 747–759 (2014).